# Coagulation factor IX analysis in bioreactor cell culture supernatant predicts quality of the purified product

Lucia F. Zacchi[1,4], Dinora Roche-Recinos [1,2,4], Cassandra L. Pegg[3], Toan K. Phung [3], Mark Napoli[2], Campbell Aitken[2], Vanessa Sandford[2], Stephen M. Mahler[1], Yih Yean Lee[2✉], Benjamin L. Schulz [1,3✉] & Christopher B. Howard[1✉]

Coagulation factor IX (FIX) is a complex post-translationally modified human serum glyco-protein and high-value biopharmaceutical. The quality of recombinant FIX (rFIX), especially complete γ-carboxylation, is critical for rFIX clinical efficacy. Bioreactor operating conditions can impact rFIX production and post-translational modifications (PTMs). With the goal of optimizing rFIX production, we developed a suite of Data Independent Acquisition Mass Spectrometry (DIA-MS) proteomics methods and used these to investigate rFIX yield, γ-carboxylation, other PTMs, and host cell proteins during bioreactor culture and after pur-ification. We detail the dynamics of site-specific PTM occupancy and structure on rFIX during production, which correlated with the efficiency of purification and the quality of the purified product. We identified new PTMs in rFIX near the GLA domain which could impact rFIX GLA-dependent purification and function. Our workflows are applicable to other biologics and expression systems, and should aid in the optimization and quality control of upstream and downstream bioprocesses.

[1] ARC Training Centre for Biopharmaceutical Innovation, Australian Institute for Bioengineering and Nanotechnology, The University of Queensland, St. Lucia, QLD, Australia. [2] CSL Limited, Parkville, VIC, Australia. [3] School of Chemistry and Molecular Biosciences, The University of Queensland, St Lucia, QLD, Australia. [4] These authors contributed equally: Lucia F. Zacchi, Dinora Roche-Recinos. ✉email: YihYean.Lee@csl.com.au; b.schulz@uq.edu.au; c.howard2@uq.edu.au

Maintaining homeostasis of the cardiovascular system requires a fine-tuned blood coagulation machinery that ensures appropriate, rapid, and localized formation of blood clots and their subsequent dissolution[1]. A key step in the coagulation pathway is the formation of the "tenase" complex, which accelerates the production of the fibrin clot. The tenase complex is formed by coagulation factors VIIIa, IXa, and X (where "a" stands for activated), which associate on the phospholipidic membrane of platelets or monocytes through their calcium-binding GLA domains[2–5] (Fig. 1). Mutations in the X-linked genes encoding factors VIII and IX lead to bleeding disorders called Hemophilia A and B, respectively[6]. Hemophilia B impacts ~1 in 25,000 male births, and is characterized by spontaneous bleeding and an inability to clot[6–8]. Factor IX (FIX) is a key player in the coagulation cascade, and FIX deficiency severely impacts the quality of life of affected individuals.

The standard therapy for Hemophilia B is prophylaxis by intravenous administration of plasma-derived or recombinant FIX (rFIX)[8]. Several rFIX products are available on the market, including BeneFIX (Pfizer, 1997), Rixubis (Baxter, 2013), Alprolix (Biogen Idec, 2014), Ixinity (Emergent Biosolutions, 2015), Idelvion (CSL, 2016), and Refixia/Rebinyn (Novo-Nordisk, 2017)[8]. These rFIX are produced in mammalian expression systems to ensure native posttranslational modifications (PTMs), which are required for rFIX activity, stability, and serum half-life[1,9]. rFIX industrial production is typically 100–1000-fold less for the same cell culture volume than monoclonal antibodies, indicating the presence of substantial biosynthetic bottlenecks[7,10]. Two well-known biosynthetic bottlenecks in rFIX are proteolysis of the propeptide and the γ-carboxylation of the GLA domain (refs. [11,12], and reviewed in ref. [10]). The variety and complexity of rFIX's PTMs, together with the low yield, make rFIX a challenging biologic to produce.

FIX is a highly posttranslationally modified glycoprotein[7,13] (Fig. 1). Immature FIX is composed of six structural domains: an N-terminal propeptide, the GLA domain, two consecutive EGF-like domains followed by a short linker, the activation peptide (AP), and the C-terminal serine protease domain (Fig. 1). Most described PTMs are on or near the GLA, the EGF-like 1, and the AP domains. The propeptide and the AP are removed through proteolysis events, and both proteolysis are required for function[1,9,14]. FIX's propeptide is cleaved by the trans-Golgi protease PACE/Furin during transit through the secretory pathway[15,16]. FIX's AP is removed in the blood by factor XIa as part of the coagulation cascade (Fig. 1), thereby activating the serine protease activity of FIX[17]. A critical PTM on FIX is γ-carboxylation of the GLA domain (Fig. 1). The GLA domain is

a 46 amino acid long region that contains 12 γ-carboxylated Glu residues in plasma-derived FIX (pdFIX)[18–20]. γ-Carboxylation is the enzymatic addition of a $CO_2$ moiety to the γ-carbon of Glu residues by the γ-glutamyl carboxylase, a vitamin K-dependent enzyme localized in the endoplasmic reticulum (ER)[4,19,21,22]. γ-Carboxylation of the GLA domain is required for calcium binding and for the formation of the tenase complex, and is thus absolutely essential for FIX function[2,3,19]. For this reason, ensuring optimal γ-carboxylation of the GLA domain is a priority for the production of rFIX. The EGF-like 1 domain is O-glucosylated at S53 (refs. [23,24]), O-fucosylated at S61 (refs. [24,25]), β-hydroxylated at D64 (refs. [26,27]), and phosphorylated at S68 (refs. [28]; Fig. 1). The linker region between the EGF-like 2 and the AP domains is O-glycosylated with O-GalNAc at S141 (refs. [29,30]; Fig. 1). The AP is a highly glycosylated 35 amino acids region with 2 N-linked glycans at N157 (refs. [30–34]) and N167 (refs. [30–34]), and O-GalNAc glycans at T159 (refs. [24,35]), T169 (refs. [24,35]), and T172 (ref. [20]; Fig. 1). In addition, the AP is sulfated at Y155 (ref. [34]) and phosphorylated at S158 (refs. [24,34]; Fig. 1). Finally the serine protease domain is N-glycosylated at N258 (ref. [30]; Fig. 1). FIX also contains 11 predicted or verified disulfide bonds, including the interchain bond between C178 and C335 (ref. [5]; Fig. 1). In addition, rFIX is O-glycosylated with O-GalNAc at T38 or T39 and T179, and phosphorylated at T159 (refs. [36,24]). The PTMs in FIX are diverse and heterogeneous, and the precise functions of most PTMs are not completely clear.

Measuring PTMs on purified biopharmaceutical proteins is critical, given the key role of PTMs in determining protein stability, half-life, function, and immunogenicity[7,37–39]. PTMs are routinely measured after product purification as part of the quality control process during manufacture. However, measuring PTMs during bioreactor operation in the bioreactor supernatant has the advantage of providing information on the dynamic biosynthetic capability of the biological system under specific bioprocess conditions. The quality of the product changes during bioreactor operation as the cellular metabolism changes due to cell aging, nutrient depletion, and toxic product accumulation[39–41]. The culture media and bioprocess operating conditions can also affect PTM occupancy and structure. Glycosylation profiles can be affected by glycoengineering strategies and the choice of cell line[42]; media composition including glucose, glutamine, lactate, ammonia, and amino acid concentrations[43–45]; and process parameters, such as dissolved oxygen, temperature, and pH[46,47]. γ-Carboxylation depends on adequate cofactors, vitamin K, and processing enzymes, and an inverse relationship has been observed between the extent of γ-carboxylation and rFIX yield[48]. Product purification can also bias the stoichiometry and structure of

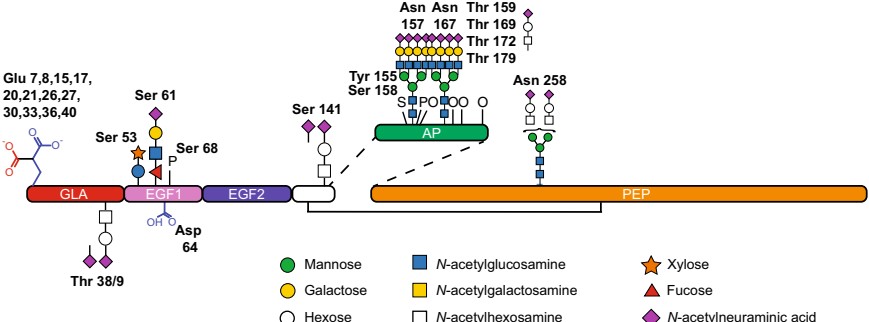

**Fig. 1 Coagulation factor IX structural domains and posttranslational modifications.** Schematic of mature factor IX (FIX) with previously described posttranslational modifications (PTMs) in plasma-derived FIX and/or recombinant FIX (figure modified from ref. [30]). The GLA domain (red) contains 12 potential sites of γ-carboxylation on E[7–40] and one O-glycan at T[38/39]. The EGF-like 1 domain (pink) contains two O-glycans on S53 and S61, β-hydroxylation of D64, and phosphorylation of S68. The EGF-like 2 domain (violet) has no known PTMs. The short domain (white) linking the EGF-like 2 domain with the activation peptide (AP, green) contains one O-glycan at S141. The AP contains two N-glycans at N157 and N167, four O-glycans at T159, T169, T172, and T179, sulfation of Y155, and phosphorylation of S158 and T159. The serine protease domain (orange) contains one N-linked glycan at N258.

PTMs in the final product, as is the case for rFIX purification which enriches for γ-carboxylated protein variants[49,50]. Measuring the product and its PTMs during bioreactor operation could therefore allow for more accurate insights, and control of the quantity and quality of the product during biosynthesis.

Mass spectrometry proteomics (MS) is a versatile tool with outstanding applicability to biopharmaceutical PTMs characterization[51]. Data independent acquisition (DIA) MS is a powerful analytical tool that allows simultaneous measurement of the relative abundance of proteins and their PTMs in complex protein mixtures[52,53]. In DIA, all peptides eluting across a liquid chromatography (LC) gradient are fragmented according to predetermined size (mass/charge) windows. By summing the abundance of preselected fragment ions of interest present in specific windows at specific retention times (RTs), it is possible to calculate the stoichiometry of modification and the relative abundance of differently posttranslationally modified peptide variants and proteins in each sample[52-58]. DIA is superior to data-dependent acquisition (DDA) workflows in that DDA fragmentation is intrinsically biased toward the more abundant peptides, while DIA allows measurement of all detectable analytes, allowing for a full exploration of the precursor landscape in a sample[58,59].

We hypothesized that using DIA-MS to estimate yield and quality of rFIX in the bioreactor supernatant would allow prediction of yield and quality of the purified product. We developed a suite of LC-MS/MS DIA workflows to measure the relative quantity of rFIX and its PTMs produced in fed-batch bioreactor cultures during bioreactor operation and after purification (Supplementary Fig. S1). We cultured Chinese hamster ovary (CHO) cells co-expressing wild-type rFIX and PACE/Furin for 13 days in two different commercial feed media (EfficientFeed A or EfficientFeed B nutritional supplements). We analyzed daily samples of clarified supernatants and POROS 50 HQ purified rFIX to characterize the PTMs on rFIX, and to measure the relative abundance of rFIX, its PTMs, and host cell proteins (HCPs) during the bioreactor operation and after purification. Our results show that analyzing bioreactor supernatant provided robust data on rFIX yield and quality that correlated with the yield and quality of the purified product. Thus, MS proteomics workflows that monitor the bioprocess protein dynamics have the potential to accelerate upstream product optimization and to aid in quality control process during the production phase.

## Results

### Performance of CHO cells expressing rFIX in fed-batch bioreactors with different feeds.
To test the hypothesis that monitoring bioreactor culture media with DIA-MS proteomics can inform on the yield and quality of the final product, we cultured CHO K1SV cells that stably co-expressed rFIX and the protease PACE/Furin in two fed-batch bioreactor conditions. The incubation conditions were identical except for the commercial feed used in each bioreactor: EfficientFeed A (bioreactor H1) or B (bioreactor H2). To obtain an overview of the physiological performance of the cultures, we measured cell viability and the concentration of key metabolites (glutamine, glutamate, ammonium, glucose, and lactate) during bioreactor operation (Fig. 2a, Supplementary Fig. S2, and Supplementary Data S1). The viable cell density (VCD) of both bioreactors reached a maximum at days 7–8 and then gradually declined (Fig. 2a). Similarly, cell viability in both bioreactors remained at ~98% until day 7, and then dropped to ~93% in H1 and to ~77% in H2, leading to the termination of the operations at day 13 (Fig. 2a). We also observed lower levels of glutamine and ammonium and a sharp increase in lactate production toward the end of the operation in bioreactor H2 compared to bioreactor H1 (Supplementary

Fig. S2). Therefore, EfficientFeed A provided better support of cellular viability and metabolism than EfficientFeed B.

Product yield is critical when considering optimization of bioreactor operational parameters. We used DIA-MS to measure the abundance of rFIX in the bioreactor culture supernatant during bioreactor operation. rFIX abundance steadily increased in both bioreactors from days 1–8 (Fig. 2b). While rFIX abundance in H1 plateaued at day 8, rFIX abundance continued to increase in bioreactor H2 (Fig. 2b). Similar results were observed by western blot (Supplementary Fig. S3). MSstats comparison of relative rFIX abundance in bioreactors H1 vs H2 showed that already at day 6 there was significantly more rFIX in the supernatant of bioreactor H2 compared to H1, and that this difference was largest at day 13 ($P < 10^{-5}$, Fig. 2c and Supplementary Data S2). Therefore, although feeding with EfficientFeed B (H2) led to lower cell viability, it increased production of rFIX in the bioreactor supernatant.

### Predicting γ-carboxylation levels in purified rFIX by analyzing the culture supernatant.
One of the key quality attributes of rFIX is γ-carboxylation of the GLA domain[7,20,60]. FIX's GLA domain contains 12 Glu residues that are essentially completely modified to γ-carboxyglutamic acid in plasma-derived FIX (pdFIX; Fig. 3a)[18-20]. Although similarly functional, CHO produced rFIX is incompletely γ-carboxylated in the last two Glu of the GLA domain[20,61,62]. Due to the critical importance of γ-carboxylation for the physiological function of FIX[60], bioreactor operation and purification procedures seek to optimize rFIX γ-carboxylation.

Fully γ-carboxylated GLA peptides are difficult to detect and identify in positive ion mode liquid chromatography electrospray ionization tandem mass spectrometry (LC-ESI-MS/MS). Some of the reasons for this include the negative charge of the carboxyl groups, that γ-carboxylation appears to hinder protease cleavage, and neutral loss of $CO_2$ upon collision-induced dissociation (CID) fragmentation[63-66]. However, uncarboxylated or partially γ-carboxylated GLA peptides can be detected in positive ion mode LC-ESI-MS/MS and used as a proxy for γ-carboxylation levels. Alternatively, γ-carboxylated peptides can be directly measured by performing methanolic derivatization of the proteins prior to proteolysis[65,66]. Methylation neutralizes the negative charge of the carboxyl groups, stabilizes the γ-carboxylation preventing neutral loss during fragmentation, and facilitates protease cleavage[65,66]. Therefore, methylation allows measurement of both incompletely and completely γ-carboxylated peptides. In addition, methylation provides site-specific information on γ-carboxylation, which is valuable for FIX because complete γ-carboxylation of GLA is not required for rFIX function[61]. Therefore, both methodologies provide complementary γ-carboxylation information.

To test how rFIX γ-carboxylation changed throughout bioreactor operation, we used DIA-MS to measure rFIX γ-carboxylation in supernatant samples from days 1 to 13 in the underivatized form, and also measured γ-carboxylation of derivatized rFIX at day 13 (Fig. 3, Supplementary Fig. S4, and Supplementary Data S3 and S4). As expected, fully γ-carboxylated GLA peptides were not detectable in the underivatized samples, except for γ-carboxylated TTE[40]FWK (Fig. 3a, Table 1, Supplementary Fig. S5a, and Supplementary Data S5). Measurement of underivatized rFIX allowed quantification of uncarboxylated ($+44 \times 0$), mono ($+44 \times 1$), and di γ-carboxylated ($+44 \times 2$) LE[7]E[8]FVQGNLE[15]R, uncarboxylated CSFE[26]E[27]ARE[30]V-FE[33]NTE[36]R, and uncarboxylated and γ-carboxylated TTE[40]FWK (Fig. 3a, b, Table 1, Supplementary Figs. S4 and S5a, and Supplementary Data S5). These results indicated that rFIX in both bioreactors was partially γ-carboxylated. The relative abundance of TTE[40]FWK carboxyforms in the supernatant of both bioreactors was similar (Fig. 3 and Supplementary Fig. S4). On the other hand, the

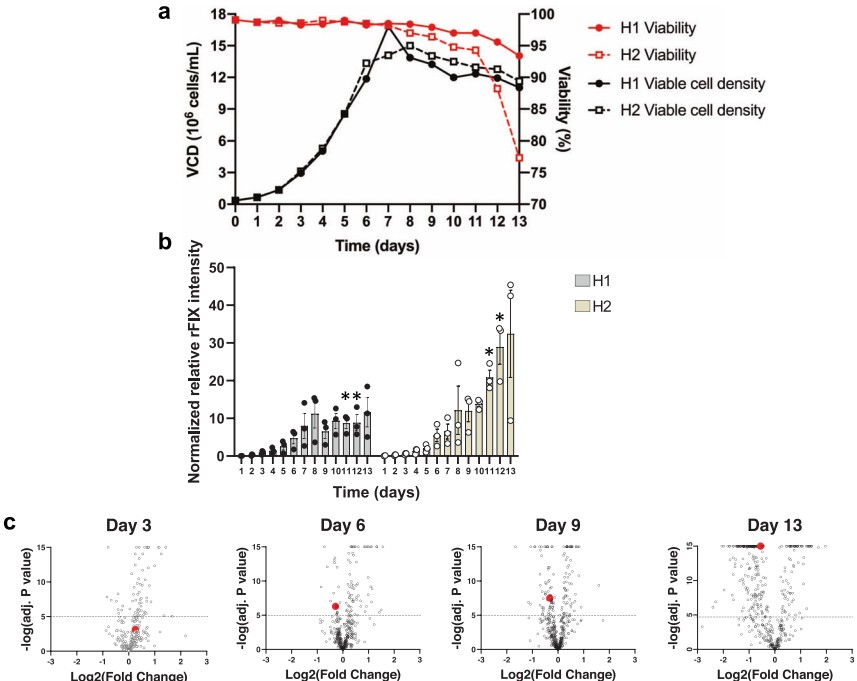

**Fig. 2 Viability and productivity of CHO cells expressing rFIX in fed-batch conditions.** CHO cells co-expressing rFIX and PACE/Furin were grown in fed-batch bioreactor mode with either EfficientyFeed A (H1) or EfficientFeed B (H2) as feeds. **a** Viability (red line) and viable cell density (VCD; black line) in H1 (solid line, closed circle) and H2 (dotted line, open square); $N = 1$. **b** Relative rFIX abundance (normalized to trypsin) in the bioreactor supernatants during operation (Mean ± SEM; multiple $t$ test, $N = 3$ independent technical replicates; $*P = 0.0072$ and $P = 0.0166$ for day 11 and day 12 in H1 vs H2, respectively, Supplementary Data S23). Individual data points are indicated in black (H1 bioreactor, gray bars) or white circles (H2 bioreactor, yellow bars). **c** Volcano plots depicting log2 of the fold change in protein abundance vs −log10 of adjusted $P$ value for comparisons of culture media of bioreactor H1 vs H2 at days 3, 6, 9, and 13. The dotted horizontal line indicates the value above which the comparisons were significant (MSstats, $P < 10^{-5}$, $N = 3$ independent technical replicates). The red dots indicate rFIX at day 3 (adjusted $P$ value = 0.00078), day 6 (adjusted $P$ value = $5.02 \times 10^{-7}$), day 9 (adjusted $P$ value = $3.1 \times 10^{-8}$), and day 13 (adjusted $P$ value = 0) in H1 vs H2. Each open circle is a unique protein.

extent of γ-carboxylation at $E^7$, $E^8$, and $E^{15}$ was lower in bioreactor H2 compared to H1, as we measured more uncarboxylated and partially γ-carboxylated $LE^7E^8FVQGNLE^{15}R$ peptides in the bioreactor H2 supernatant compared to H1, especially toward the end of bioreactor operation (days 9–13; Fig. 3b). Methylation greatly increased our ability to detect and reliably quantify γ-carboxylated rFIX, including measurement of fully γ-carboxylated variants of several GLA peptides (Fig. 3). In agreement with the measurement of underivatized rFIX, the levels of partially or uncarboxylated methylated $LE^7E^8FVQGNLE^{15}R$, $CSFE^{26}E^{27}AR$, and $E^{30}VFE^{33}NTE^{36}R$ peptides were significantly higher in bioreactor H2 supernatant compared to H1 at day 13 (Fig. 3a, c–f and Supplementary Data S4). Indeed, we detected significantly lower levels of fully γ-carboxylated methylated GLA peptides in bioreactor H2 supernatant compared to H1 (Fig. 3). Therefore, both analytical approaches unequivocally showed that rFIX in H1 bioreactor supernatant was more efficiently γ-carboxylated than in H2.

To purify rFIX we used a POROS 50 HQ strong anion exchange resin, a quaternary polyethyleneimine that binds negatively charged molecules[49,67]. POROS 50 HQ enriches for γ-carboxylated FIX due to the high negative charge of the γ-carboxyglutamic acids in the GLA domain[7,50,66]. We expected that purified rFIX from both bioreactors would be enriched in highly γ-carboxylated forms compared to the bioreactor supernatant. Indeed, the extent of γ-carboxylation in rFIX from bioreactor H1 and H2 was significantly higher after purification than in the culture supernatants (Fig. 3c–g and Supplementary Data S4g). Although rFIX in bioreactor H1 supernatant was more γ-carboxylated than in H2 (Fig. 3), we expected that purified H1 rFIX would have a similar level of γ-carboxylation as H2 rFIX. However, while purified H1 rFIX was almost completely γ-carboxylated, purified

H2 rFIX was not (Fig. 3c–g and Supplementary Fig. S4). The levels of fully γ-carboxylated GLA peptides were significantly lower in purified H2 rFIX compared to H1 (Fig. 3 and Supplementary Data S4g), and the abundance of uncarboxylated or partially γ-carboxylated GLA peptides was significantly higher in purified H2 rFIX compared to H1 (Fig. 3, Supplementary Fig. S4, and Supplementary Data S4g). These results indicated that purification reduced γ-carboxyform heterogeneity and enriched for γ-carboxylated forms in both H1 and H2 rFIX, but did not produce completely γ-carboxylated H2 rFIX (Fig. 3 and Supplementary Fig. S4). Interestingly, while γ-carboxylated peptides $LE^7E^8FVQGNLE^{15}R$, $CSFE^{26}E^{27}AR$, and $E^{30}VFE^{33}NTE^{36}R$ were enriched after purification (Fig. 3d–f), γ-carboxylated $TTE^{40}FWK$ was not strongly enriched after purification (Fig. 3g). This suggests that γ-carboxylation of $E^{40}$ is not critical for binding to POROS 50 HQ strong anion exchange resin. These results are consistent with the low conservation of the 12th GLA residue ($E^{40}$) in other γ-carboxylated coagulation factors[13], with lack of effect of loss of $E^{40}$ γ-carboxylation on FIX binding to phospholipid membrane and factor X activation[61], and with incomplete occupancy at this site on pdFIX[30]. In summary, purified H1 rFIX was more γ-carboxylated than purified H2 rFIX. It is possible that other PTMs, site-specific contributions, and differences in sample complexity (HCPs levels and variety) contributed to the purification of lower γ-carboxylated rFIX forms in H2. These results indicate that use of a strong anion exchange resin alone does not ensure that the purified rFIX will be completely modified, supporting the need for MS techniques to measure site-specific γ-carboxylation to confirm the product quality.

Altogether, we designed a positive ion mode LC-ESI-DIA-MS method to measure site-specific γ-carboxyforms with or without derivatization from the bioreactor supernatant or purified rFIX. The

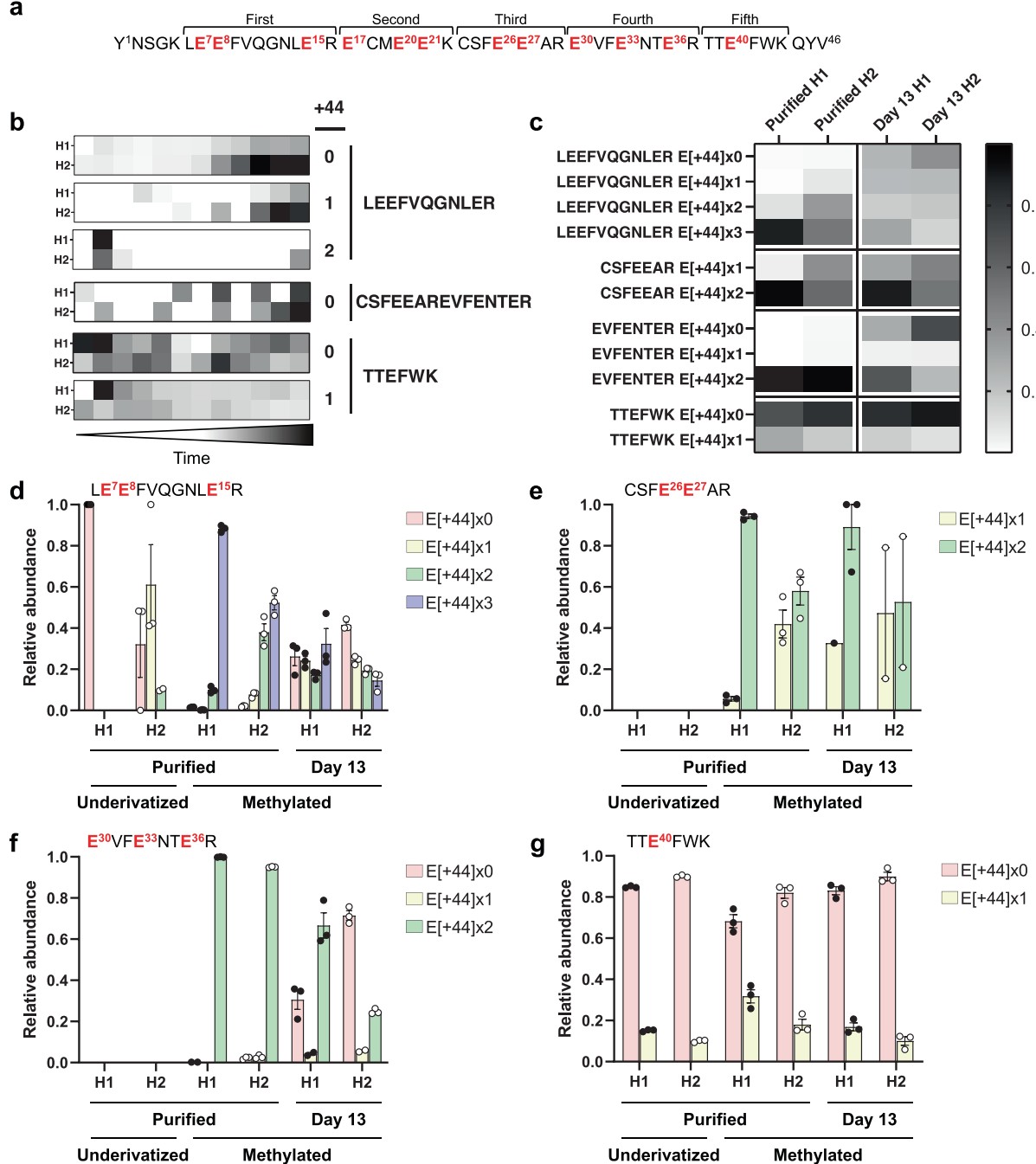

**Fig. 3 Relative abundance of γ-carboxylated GLA peptides from rFIX during bioreactor operation and after purification measured by DIA-MS.** The abundance of rFIX γ-carboxylated tryptic peptides was measured by DIA-MS in samples from bioreactor H1 and H2 supernatant and after rFIX purification. Shown are the results with and without prior derivatization by methylation. **a** Amino acid sequence of rFIX GLA domain highlighting the 12 Glu residues (in red) and the five tryptic peptides potentially measurable by LC-ESI-MS/MS. **b** Heatmaps depicting the change in mean relative abundance of unmethylated GLA γ-carboxypeptides containing 0–2 γ-carboxylations (+44) over time (13 days) in each bioreactor ($N = 2$–3 independent technical replicates). **c** Heatmaps displaying the mean relative abundance of each of the methylated γ-carboxylated peptide variants in the purified samples and in the bioreactor supernatants at day 13 ($N = 2$–3 independent technical replicates). **b**, **c** Relative abundance is depicted in a white to black scale, where white is non-abundant and black is highly abundant. **d**–**g** Bar graphs depicting the mean relative intensity of carboxyforms from purified rFIX or rFIX from bioreactor supernatant at day 13, with or without derivatization by methylation. Multiple variants of each peptide were measured by DIA-MS, containing 0–3 γ-carboxylations, 0–6 methyl groups, and two different O-glycan compositions at $T^{38/39}$ (Fig. 1). **d** $LE^7E^8FVQGNLE^{15}R$, **e** $CSFE^{26}E^{27}AR$, **f** $E^{30}VFE^{33}NTE^{36}R$, and **g** $TTE^{40}FWK$. All graphs display the mean ± SEM ($N = 1$–3 independent technical replicates). Individual data points are indicated in black (H1 bioreactor) or white circles (H2 bioreactor). Statistical comparisons (one-tailed $t$ test) for Fig. 3d–g can be found in Supplementary Data S4g.

**Table 1 Posttranslational modifications observed on pdFIX and rFIX from bioreactors H1 and H2.**

| FIX Domain | Residue(s) | Posttranslational modification | pdFIX | H1 rFIX | H2 rFIX | Refs. |
|---|---|---|---|---|---|---|
| Propeptide | $T^{-18}$VFLDHENANKIL$^{-41}$ | Proteolysis | | * | * | 15,16 |
| GLA | $E^7$, and/or $E^8$, and/or $E^{15}$ | Carboxylation | | *p | *p | 18 |
| | $E^{17}$, and/or $E^{20}$, and/or $E^{21}$ | Carboxylation | | | *p | 18 |
| | $E^{26}$ and/or $E^{27}$ | Carboxylation | | *p | *p | 18 |
| | $E^{30}$, and/or $E^{33}$, and/or $E^{36}$ | Carboxylation | | *p | *p | 18 |
| | $E^{40}$ | Carboxylation | * | *p | *p | 18 |
| | $T^{38/39}$ | O-glycan $HexNAc_1Hex_1NeuAc_1$a | | *p | *p | This study |
| | | O-glycan $HexNAc_1Hex_1NeuAc_2$a | | *p | *p | 24 |
| | | O-glycan $HexNAc_1Hex_1NeuAc_1NeuGc_1$a | | *p | *p | This study |
| | $Y^{45}$ | Sulfation/phosphorylation | | *p | *p | This study |
| EGF-like 1 | $D^{47}$ | Oxidation | | *p | *p | This study |
| | $D^{49}$ | Oxidation | *p | *p | *p | This study |
| | $S^{53}$ | O-glycan $Hex_1Xyl_1$ | * | | | 23,24 |
| | | O-glycan $Hex_1Xyl_2$ | * | *p | *p | 20,23,24 |
| | $S^{61}$ | O-glycan $Fuc_1HexNAc_1Hex_1NeuAc_1$ | * | *p | *p | 20,23,24 |
| | | O-glycan $Fuc_1HexNAc_1Hex_1NeuGc_1$ | | *p | *p | This study |
| | $S^{53}$, $S^{61}$, $S^{68}$ | O-glycan $Hex_1Xyl_2$, $Fuc_1HexNAc_1Hex_1NeuAc_1$, $Fuc_1$a | | *p | *p | This study |
| | $D^{64}$ | β-hydroxylation | *p | *p | *p | 20,26,27 |
| | $S^{68}$ | Phosphorylation | *p | | | 28 |
| EGF-like 2 | $D^{85}$ | Oxidation | *p | *p | *p | This study |
| | $D^{104}$ | Oxidation | *p | *p | *p | This study |
| | $S^{102/110}$, $T^{112}$ | O-glycans $Hex_1Xyl_2$, $Fuc_1HexNAc_1Hex_1NeuAc_1$a | | *p | *p | This study |
| Linker | $S^{141}$ | O-glycan $HexNAc_1Hex_1NeuAc_1$ | *p | *p | *p | 30 |
| | | O-glycan $HexNAc_1Hex_1NeuAc_2$ | *p | *p | *p | 29,30 |
| AP**b** | $Y^{155}$/$S^{158}$ | Sulfation/phosphorylation a | *p | *p | *p | 24,34 |
| | $N^{157}$ | N-glycan | * | * | * | 30-34 |
| | $T^{159}$ | O-glycan $HexNAc_1Hex_1NeuAc_1$a | | *p | | 24,35 |
| | | O-glycan $HexNAc_1Hex_1NeuAc_2$a | *p | *p | *p | 24,36 |
| | $N^{167}$ | N-glycan | * | * | * | 30-34 |
| | $T^{169/172}$ | O-glycan $HexNAc_1Hex_1NeuAc_1$a | *p | | | 20,24,35 |
| | | O-glycan $HexNAc_1Hex_1NeuAc_2$a | | *p | *p | 20,24,36 |
| | $T^{179}$ | O-glycan $HexNAc_1Hex_1NeuAc_1$ | | *p | *p | 20,24,36 |
| | | O-glycan $HexNAc_1Hex_1NeuAc_2$ | | *p | *p | 20,24,36 |
| | | O-glycan $HexNAc_1Hex_1NeuAc_1NeuGc_1$ | | *p | | This study |
| Protease | $D^{186}$ | Oxidation | *p | *p | | This study |
| | $D^{203}$ | Oxidation | *p | *p | *p | This study |
| | $N^{258}$ | N-glycan $HexNAc_4Hex_5NeuAc_2$ | *p | | | 30 |
| | $D^{276}$ | Oxidation | *p | *p | *p | This study |
| | $D^{292}$ | Oxidation | *p | *p | *p | This study |
| | $D^{359}$ | Oxidation | *p | *p | | This study |
| | $D^{364}$ | Oxidation | *p | *p | | This study |

*only occupied peptides identified; *p, at least one partially occupied peptide identified.
aIt was not possible to determine the precise location of the PTM.
bOccupancy inferred after enzymatic removal of N-glycans.

two approaches showed similar results, but derivatization provided a more complete analysis of site-specific rFIX quality. The results showed that the operating conditions of bioreactor H1 favored the production of lower amounts of more efficiently γ-carboxylated rFIX compared to bioreactor H2. Importantly, the results demonstrated that measuring yield and γ-carboxylation levels in the bioreactor culture supernatant could indeed predict the bioreactor conditions which led to the highest quality purified product.

**Characterization of known and new PTMs on purified rFIX.** In addition to γ-carboxylation, FIX is modified by a large number of heterogeneous PTMs, including proteolysis, N- and O-glycosylation, sulfation, phosphorylation, β-hydroxylation, and disulfide bonds[7,19] (Fig. 1). Changes in expression system or culture conditions can lead to changes in the quantity and quality of recombinant products[47,68–71], including rFIX[11,72]. Thus, we hypothesized that in addition to γ-carboxylation, the occupancy and structure of the other PTMs may also be different on rFIX produced in different bioreactor conditions. To test this, we first

performed an in-depth DDA-MS proteomic characterization of PTMs on rFIX purified from both bioreactors, and then used the information from the tryptic digests to measure PTM relative abundance both during bioreactor operation and after purification.

Digestion with a range of proteases with orthogonal activity and the addition of peptide:N-glycosidase F (PNGase F) allowed characterization of the heavily posttranslationally modified FIX domains (Fig. 1)[7,10,13]. These analyses achieved a combined coverage of ~85% of the mature rFIX protein sequence (Supplementary Fig. S6). We were able to observe most known PTMs on pdFIX and rFIX, and also new PTMs on rFIX (Figs. 1 and 4). Most new PTMs identified were only observed in rFIX (with the exception of the Asp oxidations). Most modifications showed partial occupancy in rFIX and were generally more heterogeneous than in pdFIX. The compiled data are displayed in Table 1, Fig. 4a, Supplementary Figs. S5 and S6, and Supplementary Data S5–S9. Below, we summarize the results and highlight some of our discoveries.

FIX contains a propeptide region ($T^{-18}$VFLDHENAN-KILNRPKR$^{-1}$) that is cleaved during secretion by the

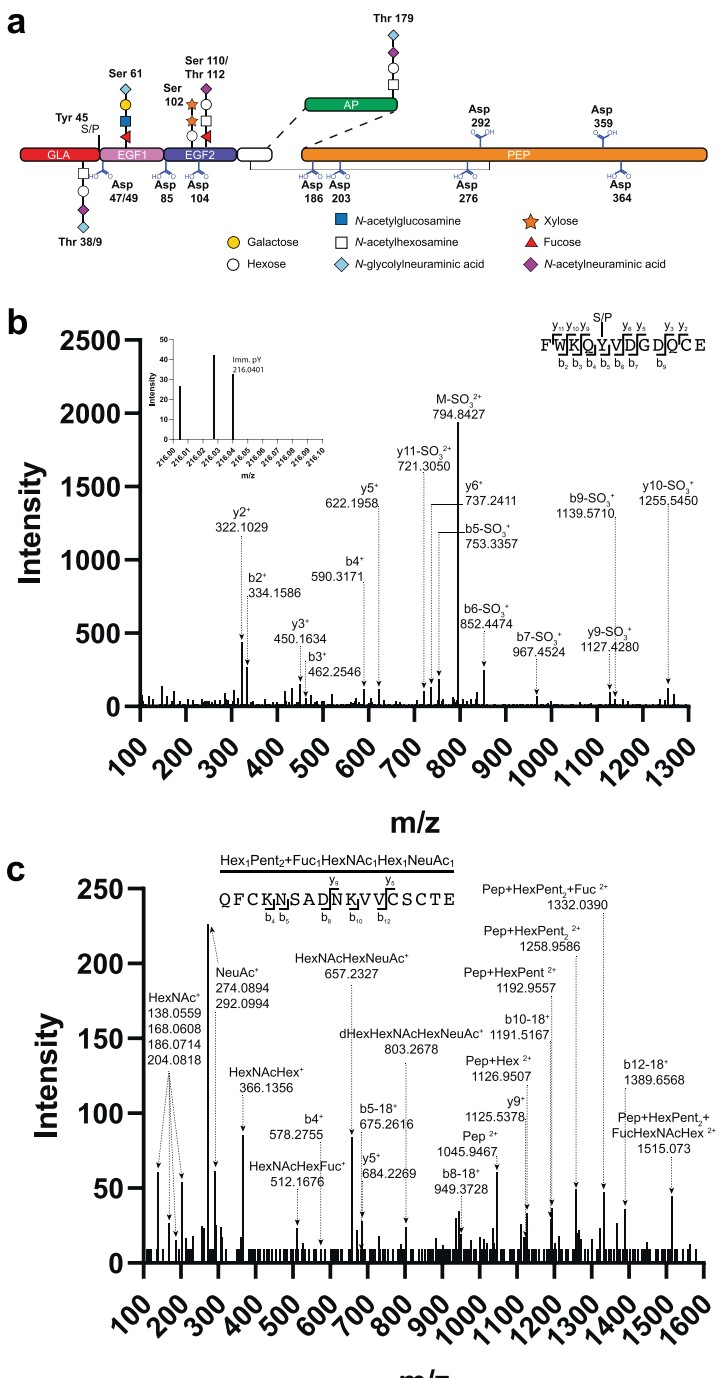

**Fig. 4 New posttranslational modifications identified on rFIX from bioreactors H1 and H2. a** Schematics of FIX containing the new PTMs identified on rFIX in this study (refer to Fig. 1 for an explanation of the colors). **a, b** CID fragmentation of select GluC rFIX peptides. **b** F$^{41}$WKQYVDGDQCE$^{54}$ peptide with sulfation/phosphorylation (S/P) at Y$^{45}$ (observed precursor $m/z$ value 834.8189$^{2+}$, Δ2.4 p.p.m.). The inset shows the phosphotyrosine immonium ion (pY, 216.0401 $m/z$, Δ2.9 p.p.m.). **c** Q$^{97}$FCKN(+1)SADN(+1)KVVCSCTE$^{113}$ glycopeptide with Hex$_1$Xyl$_2$ and Fuc$_1$HexNAc$_1$Hex$_1$NeuAc$_1$ O-glycans attached to S$^{102/110}$ and T$^{112}$ (observed precursor $m/z$ value 1107.1067$^{3+}$, Δ7.56 p.p.m.). Pep, peptide.

endoprotease PACE/Furin[15,16]. The propeptide holds a binding site for the γ-glutamyl carboxylase and is required for γ-carboxylation of the GLA domain[73–75], but failure to cleave the propeptide leads to inactive FIX[9,14]. We did not identify peptides covering the propeptide region in pdFIX, which is consistent with complete proteolytic processing of pdFIX. However, we identified peptides covering the T$^{-18}$VFLDHENANKIL$^{-6}$ propeptide region

in rFIX from both bioreactors (Table 1, Supplementary Fig. S6, and Supplementary Data S5). These results indicated that PACE/Furin activity in both bioreactor cultures was insufficient to fully process rFIX's propeptide.

We were able to obtain 87% coverage of the GLA domain in H1 and H2 rFIX and 11% in pdFIX (Supplementary Fig. S6). The low coverage observed in pdFIX is most likely due to full

γ-carboxylation of the GLA domain. We identified a variety of peptides covering the entire GLA region in both rFIX, with 0–3 γ-carboxyglutamic acids per peptide (3 being generally the maximum possible occupancy). Consistent with the DIA-MS data (Fig. 3) H2 rFIX showed higher heterogeneity of γ-carboxyglutamic acid occupancy compared to H1 rFIX (Supplementary Data S5 and Supplementary Fig. S5a). In contrast to the heterogeneity observed with rFIX, we only observed peptides containing γ-carboxylated $E^{40}$ in pdFIX (Supplementary Fig. S5a, Table 1, and Supplementary Data S5). These results are consistent with higher levels of γ-carboxylation in pdFIX compared to rFIX[20,61,62], and in H1 rFIX compared to H2 rFIX (Fig. 3, and Supplementary Figs. S4 and S5a). We observed the $T^{38/39}$ O-glycan in both H1 and H2 rFIX with the reported monosaccharide composition $HexNAc_1Hex_1NeuAc_2$ (ref. [24]; Fig. 1) and also with the new compositions $HexNAc_1Hex_1NeuAc_1$ and $HexNAc_1Hex_1NeuAc_1NeuGc_1$ (Fig. 4a, Supplementary Fig. S5a, Table 1, and Supplementary Data S5). The $T^{38/39}$ O-glycan was not detected in pdFIX, and was only observed when $E^{40}$ was not γ-carboxylated (Supplementary Data S5). Because γ-carboxylation occurs in the ER and O-glycosylation in the Golgi complex, this result suggests that γ-carboxylation of $E^{40}$ prevented O-glycosylation at $T^{38/39}$. This would also explain why this O-glycan has not been previously observed in pdFIX, since pdFIX is essentially completely γ-carboxylated[18,30]. In addition, we identified a previously undescribed sulfation/ phosphorylation event on rFIX at $Y^{45}$ (Fig. 4b, Supplementary Fig. S5b, Table 1, and Supplementary Data S5). The CID fragmentation pattern of $Y^{45}$ modified peptides showed predominantly fragment ions with neutral loss of $SO_3$, supporting the presence of a sulfation at $Y^{45}$ (ref. [76]; Fig. 4b). However, some MS/MS spectra also contained a phosphotyrosine immonium ion (216.043 m/z)[77] (Fig. 4b, inset). Other analytical workflows, including higher resolution positive ion mode or negative ion mode ESI-MS/MS analyses would be required to distinguish between sulfation or phosphorylation at $Y^{45}$ (refs. [76–78]). Analysis of the MS1 spectra of $F^{41}WKQY^{45}VDGDQCE^{54}$ peptides showed that $Y^{45}$ is largely unmodified in rFIX from both H1 and H2 bioreactors (Supplementary Fig. S5b). The $Y^{45}$ sulfation/phosphorylation was not observed in pdFIX. Overall, the GLA domain of both H1 and H2 rFIX showed higher heterogeneity than pdFIX in terms of types of PTMs, PTM composition, and occupancy. These PTMs included lower levels of γ-carboxylation and higher levels of O-glycosylation in rFIX compared to pdFIX, and the presence of a novel, to the best of our knowledge, C-terminal sulfation/phosphorylation event unique to rFIX.

We were able to obtain 100% coverage of the region containing the EGF-like 1, EGF-like 2, and linker domains in all FIX variants (Supplementary Fig. S6). The peptides covering these domains were considerably more heterogeneous in rFIX than in pdFIX, showing a larger variety of PTM types and compositions than pdFIX, and showing partial occupancy for all PTMs in rFIX (Table 1 and Supplementary Data S11). We detected all the previously known PTMs in these domains and identified several new PTMs (Figs. 1 and 4, Table 1, Supplementary Fig. S5, and Supplementary Data S5). In the EGF-like 1 domain, we identified the previously undescribed, low abundant, partial oxidation of one or both $D^{47}$ and $D^{49}$ in the peptide $F^{41}WKQYVD^{47}GD^{49}QCE^{54}$ in both H1 and H2 rFIX and in pdFIX (Fig. 4a, Supplementary Fig. S5B, Table 1, and Supplementary Data S5). In addition to the previously described $Fuc_1HexNAc_1Hex_1NeuAc_1$ O-glycan at $S^{61}$ (refs. [24,25]), we identified a minor fraction of $Fuc_1HexNAc_1Hex_1NeuGc_1$ in H1 and H2 rFIX (Fig. 4a, Table 1, and Supplementary Data S5). We also observed a peptide carrying the typical $Hex_1Pent_2$ and $Fuc_1HexNAc_1Hex_1NeuAc_1$ O-glycans, but with an additional

$Fuc_1$ on rFIX from both bioreactors (Fig. 4a). The extra fucose could be attached to one of the known O-glycans or it could be a mono O-fucosylation of $S^{68}$ (inspection of peptide Y ions did not allow for an unequivocal distinction). Mono O-fucosylation of recombinant proteins expressed in CHO, including within EGF-like domains, has been previously reported[79–82]. We observed the β-hydroxylation of $D^{64}$ in pdFIX[26,27] and in both H1 and H2 rFIX (Fig. 1, Supplementary Fig. S5c, Table 1, and Supplementary Data S5). Both H1 and H2 rFIX were more β-hydroxylated at $D^{64}$ than pdFIX, as previously reported for rFIX (Supplementary Fig. S5c)[20,83,84]. We observed the $S^{68}$ phosphorylation in pdFIX[28], but not in H1 and H2 rFIX (Fig. 4a, Supplementary Fig. S5c, Table 1, and Supplementary Data S5). No PTM has been reported so far for the EGF-like 2 domain of FIX. However, we identified low occupancy oxidations at $D^{85}$ and $D^{104}$, and two possible new sites of O-glycosylation at $S^{102/110}$ and $T^{112}$ in both H1 and H2 rFIX (Fig. 4a, c, Table 1, and Supplementary Data S5). The $D^{85}$ and $D^{104}$ oxidations were also observed in pdFIX. The $S^{102/110}$ and $T^{112}$ sites contained O-glycans with masses consistent with a $Hex_1Pent_2$ O-Glu and a $Fuc_1HexNAc_1Hex_1-NeuAc_1$ O-Fuc glycans in the $Q^{97}FCKNSADNKVVCSCTE^{113}$ peptide (Fig. 4c and Supplementary Data S5). The presence of multiple Y ions and diagnostic oxonium ions in the MS/MS spectra were also consistent with these compositions (Fig. 4c). These data suggest that the EGF-like 2 domain in rFIX contains similar O-glycans as the EGF-like 1 domain, but in lower abundance. O-glucosylation and O-fucosylation aid in EGF domain stabilization and function[81,85], which supports the presence of these glycans at this location. Overall, we were able to identify all previously reported PTMs for the EGF-like and linker domains, as well as new PTMs. With the exception of oxidation at $D^{49}$, $D^{85}$, and $D^{104}$, which were identified in both pdFIX and rFIX, all other new PTMs appeared exclusively in rFIX from H1 and H2 bioreactors: oxidation at $D^{47}$, $Fuc_1HexNAc_1Hex_1NeuGc_1$ O-Fuc glycan likely at $S^{61}$, a difucosylated $S^{53}S^{61}S^{68}$ glycopeptide, and O-glucosylation and O-fucosylation of $S^{102/110}/T^{112}$ in the EGF-like 2 domain.

We obtained a coverage of 100% of the AP in all FIX variants, and identified several known PTMs and one O-glycan composition (Fig. 1 and Supplementary Fig. S6). FIX AP is modified with four O-glycans, two N-glycans, one sulfation, and one phosphorylation (Fig. 1)[7,20,24,31,32,34,35]. The N-glycans in the AP are fucosylated or afucosylated tri- and tetra-antennary sialylated structures[31], while the O-glycans are mono and disialylated $GalNAc_1Gal_1$ structures[86]. The large size and number of glycans make characterization of the AP by LC-ESI-MS/MS difficult. To determine if the AP was glycosylated and to be able to observe other PTMs, we treated samples with PNGase F. De-N-glycosylation allowed for the detection of several peptides deamidated at $N^{157}$ or $N^{167}$ in rFIX H1 and H2, indicating that these peptides were previously N-glycosylated (Supplementary Data S5). These deamidated peptides were observed with and without other previously described modifications[20,24,35]. Partial occupancy of sulfation/phosphorylation and O-glycans on the AP has been previously described[20]. We observed the previously reported sulfation/phosphorylation of $Y^{155}/S^{158}$ (refs. [20,24,34]) in pdFIX and H1 and H2 rFIX (Fig. 1, Table 1, and Supplementary Data S5). We also observed the previously reported $HexNAc_1Hex_1NeuAc_2$ O-glycan at $S^{158}/T^{159}$ (refs. [24,35]) on pdFIX and H1 and H2 rFIX, and $HexNAc_1Hex_1NeuAc_1$ on H1 rFIX. We observed peptides containing one $HexNAc_1Hex_1NeuAc_2$ O-glycan at one of the $T^{169/172}$ sites in H2 rFIX and one $HexNAc_1Hex_1NeuAc_1$ on pdFIX (Fig. 1, Table 1, and Supplementary Data S5). Finally, we observed O-glycans with the known compositions $HexNAc_1Hex_1NeuAc_1$ and $HexNAc_1Hex_1NeuAc_2$, and the new composition $HexNAc_1Hex_1NeuAc_1NeuGc_1$ at $T^{179}$

in H1 and H2 rFIX (Fig. 4a, Table 1, and Supplementary Data S5). Overall, several known AP PTMs were identified on both pdFIX and rFIX, and one new $O$-glycan composition was identified in $T^{179}$ on rFIX. All FIX variants showed partial PTM occupancy except for the inferred completely occupied $N$-glycosites.

Finally, we obtained a coverage of 99.5%, 87.6%, and 97% of the protease domain in rFIX from H1, H2, and pdFIX, respectively (Supplementary Fig. S6). We observed the $N^{258}$ $N$-glycan with the composition $HexNAc_4Hex_5NeuAc_2$ on pdFIX[30], but not rFIX (Fig. 1, Supplementary Fig. S5e, Table 1, and Supplementary Data S5). Deglycosylation with PNGase F did not help in identifying the presence of the $N$-glycan at $N^{258}$ in rFIX because deamidation was observed in $N^{258}$ containing peptides also without enzymatic deglycosylation (Supplementary Data S5). Indeed, we note that Asn deamidation was identified throughout pdFIX and both rFIX, in Asn outside sequons and without PNGase F treatment (Supplementary Data S5). In addition, we observed several Asp oxidations with partial occupancy in pdFIX and H1 rFIX, and also in some cases in H2 rFIX, including at $D^{186}$, $D^{203}$, $D^{276}$, $D^{292}$, $D^{359}$, and $D^{364}$ (Fig. 4a, Table 1, and Supplementary Data S5). Together, our data extended the repertoire of PTMs identified in the protease domain.

In summary, we observed most of the known PTMs of pdFIX in rFIX purified from both H1 and H2 bioreactors, including γ-carboxylation, glycans, sulfation/phosphorylation, and oxidation, as well as several new PTMs (Fig. 4a). Many new PTMs were localized at or near the GLA domain, and could impact rFIX folding, function, and/or purification efficiency. Other new PTMs were localized in the EGF-like 2 domain, and may impact the stability and function of this domain. We also observed higher heterogeneity in PTMs on rFIX compared to pdFIX, both in terms of occupancy and composition. This higher PTM heterogeneity of rFIX may be the consequence of the artificial high expression CHO system combined with the changing metabolic status of cells during a fed-batch incubation, compared to a more stable native, physiological expression of pdFIX.

**Predicting the abundance of PTMs in purified rFIX by analyzing the culture supernatant.** We showed above that measuring rFIX γ-carboxylation levels in the bioreactor supernatant could predict the levels of γ-carboxylation of the purified protein (Fig. 3). While γ-carboxylation is a key PTM on FIX, other PTMs also have important functional roles and their composition and abundance can also be dependent on changes in the expression system and bioprocess parameters[3,5,9,10,13,20,26,60,85,87–92]. Therefore, it would be advantageous to be able to predict the relative abundance and composition of different types of PTMs on purified rFIX by analyzing the culture supernatant. To do this, we first identified high confidence tryptic peptides containing sites of PTM on purified rFIX and used them to build an ion library for DIA-MS (Supplementary Data S12). The ion library contained information to measure $O$-glycosylation at $T^{38/39}$, $S^{53}$, $S^{61}$, and $S^{141}$; $N$-glycosylation at $N^{258}$; and oxidation at $D^{64}$, $D^{85}$, $D^{104}$, and $D^{203}$. We used the ion library to interrogate the DIA-MS data from the purified rFIX samples and the culture supernatants from both bioreactors. To calculate the relative abundance of each PTM variant, we normalized the data for each variant to the abundance of rFIX in each sample (Supplementary Data S3). Because the DIA data were acquired with the standard 26 Da windows, the measurements for $D^{104}$ and $D^{203}$ oxidations could not be used because the unoxidized and oxidized versions of these peptides fell in the same SWATH window and the precursors eluted at a similar RT (Supplementary Data S3). This was not an issue for the $D^{64}$ variants, which eluted at different RT. We could also not quantify oxidized $D^{85}$ because of the presence of a second, unidentified peak at similar RT and in the same

SWATH window sharing a large number of the transitions selected to measure the oxidized $D^{85}$ peptide. In total, eight peptide variants were reliably quantified (FDR < 0.01), accounting for four posttranslationally modified sites (Fig. 5 and Supplementary Data S3).

We found significant differences in the abundance of different rFIX PTMs between H1 and H2 samples. There was significantly higher mono (+656) and disialylated (+947/+963) $T^{38/39}$ $O$-glycan in purified H2 rFIX compared to H1 rFIX (Fig. 5a, b). We note that the $HexNAc_1Hex_1NeuAc_2$ (+947) and $HexNAc_1Hex_1NeuAc_1-NeuGc_1$ (+963) glycoforms were quantified together, since their precursors had a similar $m/z$ ($879.8644^{2+}$ vs $887.8618^{2+}$) that fell on the same SWATH window, and eluted at similar RT (Supplementary Fig. S5a). $D^{64}$ was significantly more β-hydroxylated in purified H2 rFIX compared to H1 (Fig. 5c, d and Supplementary Fig. S5c). There was no significant difference in the abundance of unoxidized $D^{85}$ peptide in the purified rFIX from H1 and H2 bioreactors (Fig. 5e). Finally, the $S^{141}$ peptide was significantly less glycosylated in H1 rFIX compared to H2 rFIX (Fig. 5f), but the $S^{141}$ $O$-glycan was significantly more sialylated (+947) in H1 rFIX compared to H2 rFIX (Fig. 5h). We note that the $HexNAc_1Hex_1NeuAc_1$ (+656) glycosylated and unglycosylated $S^{141}$ peptides were quantified together, since their precursors had an $m/z$ that fell in the same SWATH window ($621.309^{3+}$ vs $603.3461^{2+}$ $m/z$) and eluted at similar RT (Supplementary Fig. S5d). However, the unoccupied peptide ($603.3461^{2+}$ $m/z$) had lower relative intensity than the glycosylated variant ($621.309^{3+}$ $m/z$), and the most intense ions measured for the glycopeptide were the Y ions, indicating that the signal measured for these precursors corresponds mostly to the glycopeptide. Therefore, the different feeds led to significant relative quantitative differences in glycan occupancy, glycan sialylation, and β-hydroxylation between purified H1 and H2 rFIX, and these differences were site specific.

To test if the differences in PTM abundance observed above could have been predicted by measuring those PTMs in the bioreactor, we quantified these same PTMs in the bioreactor supernatants throughout the 13 days of operation. Consistent with the purified rFIX results, we found that the abundance of the mono (+656) and disialylated (+947/+963) $T^{38/39}$ $O$-glycans was higher in the bioreactor H2 compared to H1 in the later days of the culture (Fig. 5a, b, heatmaps). There was also more β-hydroxylated $D^{64}$ in bioreactor H2 compared to H1 in the later days of the process (Fig. 5d, heatmap). Also in agreement with analysis of purified rFIX, we observed generally similar amounts of unoxidized $D^{85}$ in the H1 bioreactor and the H2 bioreactor throughout most of the 13 day process (Fig. 5e, heatmap). We were able to detect the unoccupied $S^{141}$ peptide in the earlier days during bioreactor process in both H1 and H2 bioreactors, but only weakly in the later days, suggesting that the $S^{141}$ site became occupied as the process progressed (Fig. 5f, heatmap). Unmodified $S^{141}$ peptide was still detectable in the H1 bioreactor in later days compared to the H2 bioreactor, consistent with the lower levels of occupancy of $S^{141}$ measured in purified H1 rFIX compared to H2 rFIX (Fig. 5f). The relative abundance of the two $S^{141}$ $O$-glycoforms varied throughout the process (Fig. 5g, h heatmaps). Consistent with the purified rFIX results, there was more disialylated $S^{141}$ $O$-glycopeptide in the later days of the process in bioreactor H1 compared to H2 (Fig. 5h, heatmap and Supplementary Fig. S5d). Therefore, we were able to directly measure the abundance of several rFIX PTMs in bioreactor supernatant samples, and these site-specific PTM profiles were generally consistent with purified rFIX.

We observed that some peptide variants were measurable even in the earlier days of the bioreactor process, when rFIX concentration was low, while other peptide variants were difficult to measure even in the later days of the bioprocess. This was especially the case in the H1 bioreactor, in which rFIX relative

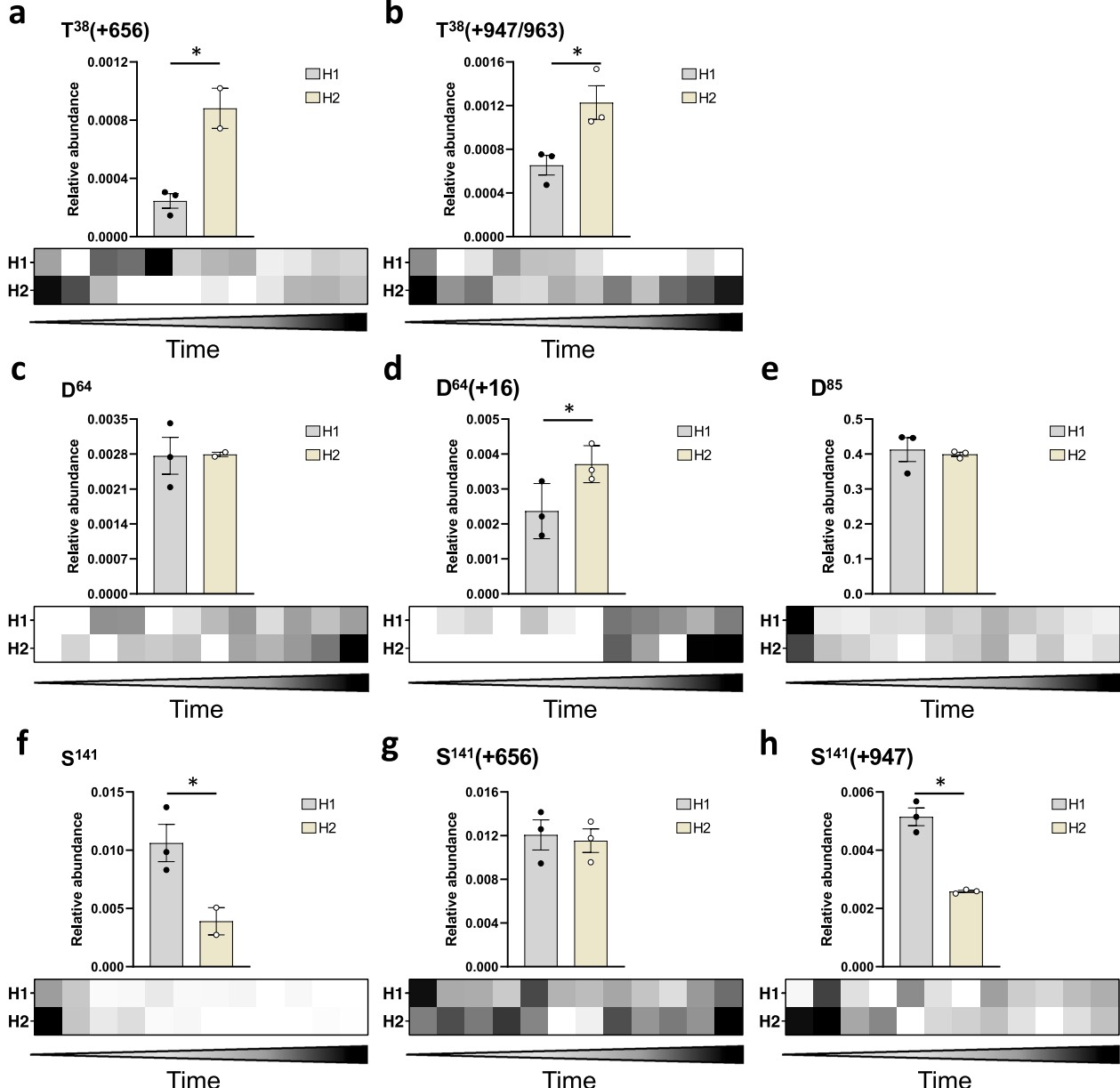

**Fig. 5 Relative abundance of posttranslational modifications on rFIX during bioreactor operation and after purification measured by DIA-MS.** Bar graphs depict peptide relative abundance in purified samples from H1 (gray) or H2 (yellow) bioreactor. The heatmaps show peptide relative abundance in each bioreactor supernatant over time (days 1–13), with white representing low abundance and black high abundance. Values correspond to the mean ± SEM (bar graphs) or the mean (heatmaps) ($N = 2$–$3$ independent technical replicates) of the normalized abundance of each posttranslationally modified rFIX peptide relative to the abundance of rFIX in each sample. One-tailed $t$ test: $*P < 0.05$. Individual data points are indicated in black (H1 bioreactor) or white circles (H2 bioreactor). Shown are: **a** $T^{38/39}$ O-glycan HexNAc$_1$Hex$_1$NeuAc$_1$ (+656) ($P = 0.00678$). **b** $T^{38/39}$ O-glycan HexNAc$_1$Hex$_1$NeuAc$_2$ (+947) and HexNAc$_1$Hex$_1$NeuAc$_1$NeuGc$_1$ (+963) ($P = 0.0163$). **c** $D^{64}$ with no oxidation ($P = 0.478$). **d** $D^{64}$ β-hydroxylation (+16) ($P = 0.0353$). **e** $D^{85}$ with no oxidation ($P = 0.358$). **f** $S^{141}$ unglycosylated ($P = 0.0289$). **g** $S^{141}$ O-glycan HexNAc$_1$Hex$_1$NeuAc$_1$ (+656) and low abundant $S^{141}$ unglycosylated peptide ($P = 0.389$). **h** $S^{141}$ O-glycan HexNAc$_1$Hex$_1$NeuAc$_2$ (+947) ($P = 0.0006$).

abundance was lower than in the H2 bioreactor (although the number and variety of HCPs was also lower in H1 bioreactor compared to H2). Even though GluC provided a better coverage of rFIX PTMs (Supplementary Data S5), we performed the quantification studies using trypsin, a standard choice in the field due to its robustness and frequency of cleavage sites. However, future studies may benefit from the use of GluC for quantification of rFIX PTMs[93].

In summary, using DIA-MS to measure rFIX PTMs in the bioreactor supernatant and after purification, and consistent with the results for the γ-carboxylation measurements (Fig. 3), we

observed a strong correlation between the relative abundance of the rFIX's PTMs measured in the bioreactor supernatant and in the purified material (Fig. 5 and Supplementary Fig. S4). Together, these data show that measurements in the bioreactor can predict measurements in the purified forms.

**Predicting purity of purified rFIX by measuring changes in host cell proteins during bioreactor operation.** HCPs present in the bioreactor supernatant are secretory and intracellular proteins released into the culture medium by the cells through secretion

and cell lysis. A decrease in cellular viability is usually accompanied by an increase in the diversity and abundance of HCPs in the culture supernatant[94,95]. Bioreactor H2 showed greater loss of viability compared to H1 during the last days of the bioprocess (Fig. 2a). Thus, we predicted that the HCP content in H2 culture supernatant would be higher than in H1. To test this idea, we used DIA-MS to measure changes in HCP abundance in the culture supernatant. Using a combined ion library (Supplementary Data S13) containing 768 proteins identified in the bioreactor culture supernatants (global FDR < 0.01), we confidently measured the abundance of 694 of these proteins throughout the process (Supplementary Data S13–15). We observed that proteome complexity and relative abundance of most proteins increased through time in both bioreactors, as expected since proteins accumulate in the culture supernatant in a fed-batch bioreactor (Supplementary Fig. S7 and Supplementary Data S15). To compare the proteomic differences between both bioreactors, we performed MSstats analysis on the DIA-MS data at different time points throughout the process (days 3, 6, 9, and 13; Fig. 2c and Supplementary Data S2). We confidently measured an increasing number of proteins in both bioreactors as the processes progressed, from 293 total proteins at day 3 to 458 total proteins at day 13 (FDR < 0.01; Figs. 2c and 6, Supplementary Fig. S7, and Supplementary Data S2). Most proteins showed no significant difference in abundance between bioreactors H1 and H2 up to day 9 (Fig. 6, 225–322 proteins or ~70–76%, $P > 10^{-5}$). However, at day 13 most of the proteins measured were significantly more abundant in the H2 bioreactor compared to H1 (203 proteins, 44.3%, $P < 10^{-5}$, Fig. 6 and Supplementary Data S1). Gene Ontology analysis of the most abundant proteins in bioreactor H2 culture supernatant at day 13 showed that these proteins were associated with the cytoskeleton, the secretory pathway, the lysosome and proteasome (proteases, hydrolases, isomerases, among others), metabolism (glycolysis and amino acid biosynthesis), and apoptosis (Supplementary Data S2 and S16). Therefore, there were large proteomic changes throughout bioreactor operation associated with using the different EfficientFeeds, which exacerbated as the cultures progressed. The more dramatic proteomic and metabolic changes observed in

bioreactor H2 began at approximately day 11 (Figs. 2 and 6, and Supplementary Figs. S2 and S7), when the batches were no longer being fed (last feed was at day 10). Together, these results suggest that unmet metabolic demands in bioreactor H2 led to the higher loss of viability, which in turn led to the higher HCP release into the supernatant in bioreactor H2 compared to H1.

HCPs in the bioreactor supernatant can impact the purification of recombinant products and the quality of the final product[96,97]. Because of the more efficient metabolism, higher cell viability, and lower complexity of HCPs in the supernatant of bioreactor H1 (Figs. 2 and 6), we predicted that the purified H1 sample would have lower co-purifying HCP complexity and abundance relative to rFIX compared to H2. To test if the H1 purified sample had lower HCP complexity than the H2 purified sample, we performed independent DDA searches of the purified H1 and H2 rFIX samples (Supplementary Data S17 and S18). As expected, we confidently identified 18 co-purifying HCPs in the H1 sample and 51 HCPs in the H2 sample (1% global FDR) (Supplementary Table S1, and Supplementary Data S11 and S19). Most of the proteins co-purifying with H1 rFIX also co-purified with H2 rFIX (13 proteins; Supplementary Table S1). Except for histone H2B, all the other 12 common protein contaminants were secretory proteins or intracellular proteins that can be potentially secreted, such as carboxypeptidase, lipoprotein lipase (LPL), clusterin, BiP, thrombospondin-1, vitamin K-dependent protein (S), procollagen C-endopeptidase enhancer 1, serine protease HTRA1, HSP90α (ref. [98]), peroxiredoxin[99], and thioredoxin reductase 1 (ref. [100]; Supplementary Table S1). Several of these proteins bind calcium, including carboxypeptidase, LPL, peroxiredoxin, thrombospondin-1, and vitamin K-dependent protein S, which contributes to explaining their co-purification with rFIX in our system (Supplementary Table S1). Other proteins like BiP, serine protease HTRA1, and histones are common CHO HCP contaminants observed during downstream recombinant product purification[101]. Thirty-five additional proteins co-purified only with H2 rFIX, most of which were intracellular proteins (23 out of 35 proteins, ~66%, Supplementary Table S1 and Supplementary Data S19). Therefore, purification using POROS 50 HQ resin enriched for rFIX (Supplementary Fig. S8 and Supplementary

| Day | More abundant in H1 | More abundant in H2 | Not significantly different |
|---|---|---|---|
| 3 | 59 (20.1%) | 9 (3.1%) | 225 (76.8%) |
| 6 | 68 (18.4%) | 20 (5.4%) | 282 (76.2%) |
| 9 | 65 (14.3%) | 67 (14.8%) | 322 (70.9%) |
| 13 | 79 (17.2%) | 203 (44.3%) | 176 (38.4%) |

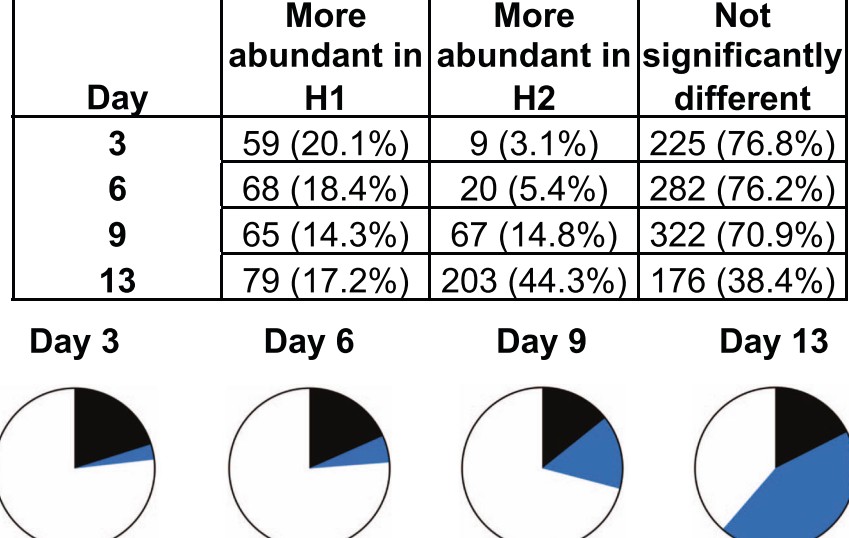

**Fig. 6 DIA-MS relative quantification of host cell proteins in the supernatant during bioreactor operation.** The table and pie charts show the number (and %) of proteins that were either significantly more abundant in H1 (black), significantly more abundant in H2 (blue), or not significantly different (white) at days 3, 6, 9, and 13 of the bioreactor process (MSStats, $P < 10^{-5}$, N = 3 independent technical replicates).

Data S2), but additional proteins co-purified with rFIX in both samples, with the H2 sample being more complex. The increased complexity of the purified H2 sample, and the predominant presence of intracellular proteins and chaperones were consistent with increased cellular stress and cell lysis in bioreactor H2 compared to H1 (Figs. 2 and 6, Supplementary Fig. S7, and Supplementary Table S1). Thus, in agreement with our hypothesis, the lower complexity of HCPs in the bioreactor supernatant correlated with lower complexity of HCPs after purification.

To test the idea that the lower abundance of HCPs in bioreactor supernatants would lead to the higher purity of rFIX relative to HCPs after purification, we performed DIA-MS of the purified samples (Supplementary Data S15). As expected, rFIX was significantly more abundant after purification in the H1 sample compared to the H2 sample ($P < 10^{-5}$, Fig. 7, box, red bar, Supplementary Data S2). This result was validated by nonreducing sodium dodecyl sulfate–polyacrylamide gel electrophoresis (SDS–PAGE) of purified rFIX samples (Supplementary Fig. S8). This result is interesting because rFIX was less abundant relative to HCPs in the H1 bioreactor supernatant compared to the H2 bioreactor, indicating a differential enrichment during purification (Figs. 2b and 7, blue bar). This differential enrichment was likely driven by a combination of the higher γ-carboxylation levels in H1 rFIX compared to H2 and the lower HCP contaminant complexity in the H1 supernatant compared to H2. There were 12 other proteins with significantly different abundance between both bioreactors ($P < 10^{-5}$; Fig. 7 and Supplementary Data S2). We observed significantly more thrombospondin-1 and carboxypeptidase in the purified H1 rFIX sample compared to the purified H2 rFIX sample (Fig. 7, red bars, Supplementary Table S1, and Supplementary Data S2). Thrombospondin-1 and carboxypeptidase are calcium binders, and were also more abundant in the supernatant from the H1 bioreactor compared to the H2 bioreactor at day 13 (Fig. 7, blue bars). Conversely, we observed significantly more histone H2B, histone H4, BiP, peroxiredoxin-1, clusterin, glutamine synthetase, translation initiation factor eIF3c, ADE2, and nucleolar phosphoprotein p130 in the purified H2 rFIX sample compared to the purified H1 rFIX sample (Fig. 7, red bars, and Supplementary Data S2 and S13, in italics). Most of these proteins do not bind calcium, and it is possible that the increased load of intracellular proteins in bioreactor H2 interfered with rFIX purification,

leading to the enrichment of non-calcium-binding intracellular proteins. Thus, as predicted, H1 bioreactor process conditions led to a higher relative yield of purified rFIX and lower relative abundance and complexity of HCPs than H2 conditions.

Altogether, our results indicated that bioreactor H1 outperformed H2 in terms of metabolic and cell viability profile, and in terms of rFIX quality (γ-carboxylation and protein co-contaminants) and purification efficiency. We were not only able to analyze the abundance and complexity of the proteome as it shifted during the bioprocess and after purification, but we were also able to monitor the abundance of diverse PTMs with site specificity, including the critically important γ-carboxylation of the GLA domain, and showed that their relative abundance in the supernatant correlated with their relative abundance postpurification. Collectively, the results indicate that a detailed DIA-MS analysis of the culture supernatant during bioreactor operation can predict the quality and yield of the purified product.

## Discussion

rFIX is commercialized as a replacement therapy for Hemophilia B (refs. [8,10]). Considerable effort has been devoted to optimize rFIX bioprocess and expression systems[7,10–12,102–106]. FIX is highly post-translationally modified with a variety of heterogeneous PTMs, and GLA domain γ-carboxylation is the most functionally relevant PTM[13,18–20]. Changes in bioprocess operation conditions can lead to changes in occupancy and composition of PTMs, including the level of GLA γ-carboxylation[8,10–13,49,62,72,88–91,102,103,106–108]. As we show here, assessment of rFIX yield and quality post-purification cannot provide a complete picture of the bioreactor performance, because rFIX purification enriches for highly γ-carboxylated product. A more effective strategy to analyze bioreactor performance is to measure yield and PTM abundance and heterogeneity in the bioreactor supernatant. To do this, we developed and implemented a suite of DIA-MS proteomics workflows to characterize, and accurately monitor rFIX yield and quality in the bioreactor supernatant. We applied these workflows to compare two 13-days fed-batch bioprocesses which only differed in the chemical composition of the feeds. We observed large differences in rFIX yield, and in the type and relative quantity of rFIX PTMs between both fed batches, including γ-carboxylation. Our results indicate that measurements of rFIX quality and quantity during bioreactor operation provide a

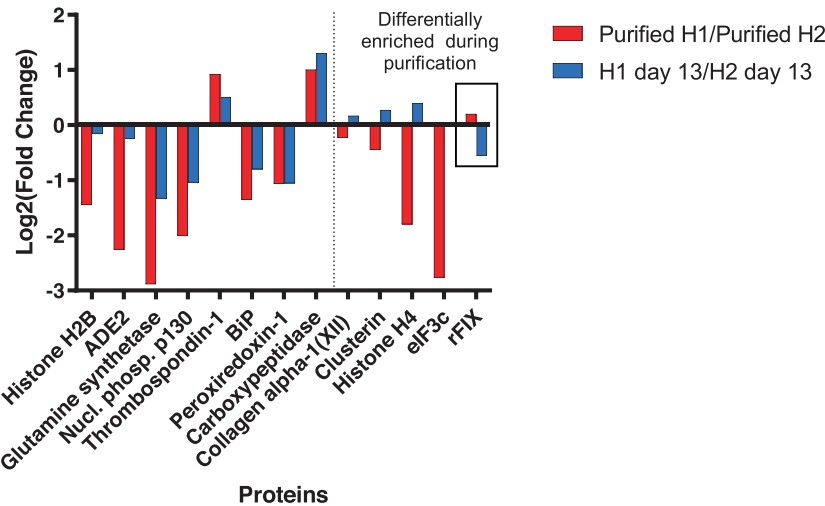

**Fig. 7 Comparison of changes in relative abundance of rFIX and co-purifying HCPs before and after purification.** Comparison of the changes in relative abundance (log2(fold change)) of the 13 proteins that were significantly different in purified H1 vs H2 samples before and after purification (H1 vs H2 (red), and H1 at day 13 vs H2 at day 13 (blue); MSstats, $P < 10^{-5}$, $N = 3$ independent technical replicates). rFIX comparison is highlighted with a box.

real-time assessment of culture performance, and allow an accurate estimation of post-purification results.

The most critical PTM on FIX, and also one of the most challenging to observe by positive ion mode LC-MS/MS, is γ-carboxylation[63–66]. Here, we show for the first time that DIA-MS is an excellent tool to measure rFIX γ-carboxylation both in the bioreactor supernatant and after purification. To measure rFIX γ-carboxylation, we used two complementary techniques which showed similar results: direct measurement of uncarboxylated or partially γ-carboxylated GLA peptides and measurement of all GLA peptide γ-carboxyforms after methanolic derivatization (Fig. 3 and Supplementary Fig. S4). Direct GLA peptide measurement in positive ion mode can provide an accurate estimation of rFIX uncarboxylation levels[64]. However, derivatization provided site-specific γ-carboxylation information and allowed for monitoring of the fully γ-carboxylated peptides, giving a more comprehensive overview of rFIX γ-carboxylation. The results of underivatized and derivatized carboxypeptide measurements were consistent and emphasized critical differences in the quality of rFIX produced in both fed batches (Fig. 3 and Supplementary Fig. S4). The γ-carboxylation measurements also demonstrated that the purification strategy was unable to distinguish between fully and substantially uncarboxylated GLA rFIX, highlighting a weakness of the enrichment procedure. It is possible that other PTMs on the GLA and EGF-like domains (such as the O-glycans at $T^{38/39}$, the sulfation/phosphorylation at $Y^{45}$, and the oxidation at $D^{47/49}$, Fig. 4 and Table 1) could impact the conformation of the GLA domain and EGF-like domains or interact with the POROS 50 HQ resin, facilitating purification of partially γ-carboxylated species. For example, H2 rFIX showed significantly higher sialylation levels at $T^{38/39}$ compared to H1 rFIX (Fig. 5), and these additional negative charges at the end of the GLA domain could be partially responsible for the purification of incompletely γ-carboxylated H2 rFIX. Importantly, our workflows can detect the onset of measurable differences in GLA γ-carboxylation (and other PTMs) during bioreactor operation (Fig. 3), and this data can be correlated with bioreactor metabolic performance (Supplementary Fig. S2), and relative FIX yield (Fig. 2c). While we performed all of our experiments in positive ion mode, negative ion mode DIA-MS could be an interesting alternative to identify and quantify highly negatively charged precursors, such as FIX GLA peptides[109,110]. Together, we demonstrate that DIA-positive ion mode MS provided an accurate depiction of rFIX γ-carboxylation levels both in the supernatant and after purification, and that these workflows can greatly aid in optimization and quality control procedures for rFIX production.

Similar to other coagulation factors, FIX is modified by a large number of PTMs (Fig. 1) that impact its biosynthetic efficiency, function, and/or pharmacokinetic properties[3,5,9,10,13,19,20,26,60,85,87–92]. For example, rFIX N-linked glycans (native and engineered) influence rFIX in vivo half-life[87,111–114] and may modulate rFIX activation[31], while rFIX EGF-like 1 O-glycans stabilize the domain and likely participate in protein–protein interactions[85,92]. Changes in the expression system and incubation conditions alter the number and variety of PTMs in recombinant proteins[7,10,11,20,62,102,107], and new, unexpected PTMs may appear. Indeed, through a detailed MS proteomic analysis, we identified new PTMs on purified rFIX, including a sulfation/phosphorylation on $Y^{45}$, several Asp oxidations spread throughout the molecule, and new O-glycan sites and compositions (Fig. 4 and Table 1). The new $Y^{45}$ modification was located within the $F^{41}WXXY^{45}$ aromatic amino acid stack at the end of the GLA domain, which

is conserved in factors VII, IX, and X, protein C, and prothrombin[19]. Studies on prothrombin showed that the hydrophobic stack contributes to the $Ca^{2+}$ induced folding of the GLA domain and protects the $C^{18}$-$C^{23}$ disulfide bond from solvent exposure (in prothrombin the residues are $C^{17}$–$C^{22}$ and $Y^{44}$)[115]. Tyr O-sulfation is performed by the trans-Golgi network tyrosylprotein sulfotransferase[116], while Tyr phosphorylation is performed by tyrosine kinases located throughout the secretory pathway[117]. Therefore, the impact of the $Y^{45}$ modification on GLA folding would likely depend on whether the $Y^{45}$ modification occurs before or after the folding of the GLA domain. In addition, the hydrophobic stack may interact with phospholipidic membranes[19], helping the formation of the tenase complex. Therefore, the presence of a sulfation/phosphorylation event on $Y^{45}$ may affect important roles of the GLA aromatic stack and FIX function. Mature FIX contains 18 Asp residues, and $D^{64}$ is β-hydroxylated[26,27]. We identified 10 additional oxidized Asp in rFIX (11 and 9 total oxidized Asp in rFIX from bioreactor H1 and H2, respectively), and almost all of these were also identified in pdFIX (Fig. 4 and Table 1). None of these additional Asp oxidations had been previously reported for pdFIX. All oxidations showed partial occupancy and, with the exception of $D^{64}$ β-hydroxylation, they were in lower abundance compared to the non-oxidized peptide (Supplementary Fig. S5b, c). It is possible that these modifications are an artifact of sample processing[118,119]. Oxidative artifacts have been described on Met and Cys (both sulfur-containing amino acids), and Trp and His, but not on Asp (refs. [118–121] and www.unimod.org). Alternatively, the modifications may have been incorporated during the bioprocess, either in the cells or in the bioreactor[122], and they may be physiologically relevant to FIX's activity. Finally, our glycan analyses of purified rFIX revealed a portion of O-glycans decorated with NeuGc (Fig. 4, Supplementary Fig. S5a, and Table 1), a nonhuman type of sialic acid that is undesirable for biotherapeutic production due to its negative impact on efficacy and safety[37]. We observed a higher level of structural heterogeneity in rFIX O-glycans compared to pdFIX (Table 1), which is expected for product from a fed-batch process. We also identified potential new O-glycan sites, including the intriguing presence of an O-Glu and O-Fuc in the EGF-like 2 domain (Fig. 4c, Table 1, and Supplementary Data S6–S9). These EGF-like 2 glycans would not be attached to Ser/Thr within the known O-glucosylation $C^1XSX(P/A)C^2$ (refs. [23,123]) or O-fucosylation $C^2XXXX(S/T)C^3$ (refs. [81,124,125]) consensus sequences (the superscript indicates the Cys residue within the EGF-like domain). However, recent studies uncovered new O-glucosyltransferases that can O-glucosylate Ser residues between the third and fourth Cys of EGF-like domains[126], precisely where we map the new EGF-like 2 $Hex_1Pent_2$ O-glycan in rFIX. In fact, human coagulation factor X is O-glucosylated at $S^{106}$ in the EGF-like 2 domain, also between the third and fourth Cys residues and in a position equivalent to the $S^{102}$ residue in rFIX[127]. These O-glycans could be attached by yet uncharacterized O-glycosyltransferases or by known enzymes with different substrate and/or sugar donor specificity able to glycosylate with low efficiency a wider range of substrates, as shown for the POGLUT1 O-glucosyltransferase, responsible for O-glucosylation at $C^1XSX(P/A)C^2$ (refs. [23,123,128]). Alternatively, O-glycans at EGF-like 2 domain may negatively impact rFIX biosynthesis, leading to low occupancy of these sites in the secreted molecule. The O-glucosylation and O-fucosylation of EGF-like domains are important for EGF domain stability and the interaction with binding partners[80,85,92], which supports their presence also in the EGF-

like 2 domain of rFIX. Altogether, our results demonstrate that a thorough analysis of the composition and abundance of all measurable PTMs in rFIX is required to confidently assess the impact of optimization procedures and for quality control purposes.

One of the highlights of this study was the development of DIA-MS workflows to measure rFIX yield and PTMs in bioreactor supernatant (Supplementary Fig. S1). Measuring product quantity and quality in the bioreactor is important because purification of rFIX enriches for γ-carboxylated rFIX, thereby altering the relative abundance of γ-carboxylated rFIX, and potentially also of other PTMs, in the samples. Post-purification analysis may not always be representative of the metabolic and biosynthetic capabilities of the expression system and bioprocess parameters utilized. We show in Fig. 7, a clear example of how post-purification analysis did not correlate with bioreactor performance: the relative abundance of rFIX in bioreactor H2 was significantly higher than in bioreactor H1 at the end of the fed batch (Figs. 2b and 7), but there was more purified H1 rFIX relative to HCPs than purified H2 rFIX (Figs. 2c and 7). This is likely because H2 rFIX was also significantly less γ-carboxylated than H1 rFIX (Fig. 3 and Supplementary Fig. S4), and because there was a larger variety of contaminating HCPs in the H2 supernatant compared to H1 (Supplementary Fig. S7 and Supplementary Data S2). If purification efficiency was the only measure of rFIX yield and quality available, we could only conclude that H1 bioreactor conditions allowed for the production of more γ-carboxylated rFIX, but we would have no understanding of how the feeds impacted rFIX biosynthesis as a whole. Even more, this conclusion would be based on the false assumption that only highly γ-carboxylated species were purified with POROS 50 HQ (Fig. 3). However, we showed here that a substantial proportion of purified H2 rFIX was not highly γ-carboxylated (Fig. 3). Without data from the bioreactor supernatant describing rFIX yield, HCP complexity, and the critically detailed site-specific analysis of γ-carboxylation and PTMs in rFIX, our ability to optimize bioreactor operation conditions would be limited and biased.

As the world population increases and ages, there is a greater need to lower the cost of biotherapeutic production. Medical and biotechnological advances have also led to an increase in the number of biotherapeutics with numerous diverse PTMs in the production pipeline. Many PTMs require complex biosynthetic pathways, leading to lower yields. The biopharmaceutical industry faces the challenge of overcoming biosynthetic bottlenecks while optimizing expression systems and bioprocessing parameters to make more of a better product, under time pressure. The DIA-MS workflows developed here are specifically designed to assist in the assessment of bioreactor performance to accelerate identification of factors impacting biosynthetic capability and capacity. These workflows are versatile and can be modified to suit the monitoring of any biologic of choice for optimization and quality control purposes. For example, incorporating stable-isotope labeled standards during the sample preparation would allow absolute quantification of the biopharmaceutical product before and after purification[129]. Improvements in the sensitivity, speed, and automation of MS instrumentation and sample preparation may also facilitate the application of these workflows for real-time industrial upstream and downstream bioprocess and product monitoring.

## Methods

**Mammalian cell line, seed train, and fed-batch conditions**. We used a CHO K1SV cell line stably expressing a modified version of human coagulation factor IX (rFIX, accession number P00740 in UniProtKB (www.uniprot.org), with Q²G and P⁴⁴V amino acid substitutions) integrated using the glutamine synthetase

expression system[130], and the protease PACE/Furin (accession number P09958, UniProtKB; provided by CSL, Marburg, Germany).

Culture flasks were inoculated at a cell density of $0.3 \times 10^6$ cells/mL, and passaged every 3 days when VCD reached $2.5 \times 10^6$ cells/mL in chemically defined protein-free medium CDCHO (Invitrogen) supplemented with 25 μM methionine sulfoximine and 50 μg/L reduced menadione sodium bisulfite (rMSB, vitamin K, required to support rFIX production) in a 1 L vented cap Erlenmeyer shake flask with a working volume of 300 mL (Corning, USA). Cells were incubated in an orbital shaking incubator (Kuhner Shaker) at 120 r.p.m., 37 °C, 5.0% $CO_2$, and 70% humidity. Two 5 L bioreactors (Sartorius) were seeded at $0.3 \times 10^6$ cells/mL in 3 L (total working volume) of CDCHO medium (Invitrogen) supplemented with 50 μg/L rMSB. Bioreactors were fed CHO CD EfficientFeed A (A1023401, Invitrogen, called bioreactor H1) or EfficientFeed B (A1024001, Invitrogen, bioreactor H2) nutrient supplements as a daily bolus starting on day 3 until 10, up to the equivalent of 40% (1.2 L) of the total postinoculation working volume of the bioreactor, following manufacturer's instructions. Two downflow segmented tri-blade impellers provided mixing and gas bubble dispersing at 200 r.p.m. The bioreactors were maintained at a pH of $7.1 \pm 0.3$, 37 °C, and 40% dissolved oxygen (controlled by constant headspace (150 mL/min) and air sparging (9 mL/min) with additional oxygen sparging on demand). Glucose was fed on demand to maintain concentration between 3–6 g/L. Cultures were terminated when cell viability in at least one bioreactor reached <80%.

**Sample collection and metabolite and physical measurements**. The bioreactor was sampled daily for off-line measurements. Total cell density, cell viability, and cell size were measured using a cell counter (Vi-cell Beckman Coulter, Fullerton, CA). Glucose, lactate, ammonium, glutamine, and other metabolites were measured with a NOVA Flex BioProfile analyzer (NOVA biomedical). pH was measured using Seven Excellence Multiparameter (METTLER TOLEDO). Oxygen (pO₂) and carbon dioxide (pCO₂) partial pressure were measured using a Siemens Blood Gas Analyzer RapidLab 248. Technical replicate samples of 15 mL were collected daily from pre-inoculation day to day 13, centrifuged at 2000 r.c.f. for 10 min, filtered through a 0.2 μm polyethersulfone filter (Pall), and stored at −80 °C in 500 μL aliquots in matrix tubes (Thermo Fisher) for further analysis. Technical replicate bulk sampling of 100 mL aliquots was performed from day 5 to day 13. On termination day (day 13), 2 L of the total working volume was collected and centrifuged at 2000 r.c.f. for 60 min, filtered through a 0.2 μm polyethersulfone filter (Pall), and stored at −80 °C.

**Anion exchange chromatography**. rFIX was purified by ÄKTA Pure (GE Healthcare) chromatography controlled with UNICORN 7.0 software. A total of 1.6 L of frozen clarified culture media from both bioreactors was thawed in a 25 °C water bath. The initial conductivity of the media was measured and EDTA levels were adjusted to 35 mM for a final sample conductivity of 15–16 mS/cm, by either adding 0.75 M EDTA and 1.32 M Tris base solution if the conductivity was <13.5 mS/cm, or by diluting the sample with 20 mM Tris HCl buffer (20 mM Tris HCl buffer pH 7.02, conductivity 1801 μS/cm at 21 °C) if the conductivity was >13.5 mS/cm. Anion exchange chromatography was performed as previously described[50]. Briefly, two prepacked columns with POROS 50 HQ resin (0.8 cm ID × 5.0 cm $H$, $V = 2.5$ mL, strong anion exchange resin, Repligen GmbH) were first equilibrated with five column volumes (CV) of EDTA equilibration buffer (50 mM MES buffer pH 5.02, 100 mM NaCl, 50 mM EDTA, conductivity 15.90 mS/cm at 21.8 °C). The clarified media sample from each bioreactor was applied to each column. Unbound proteins were washed off with five CV of NaCl wash buffer (50 mM MES buffer pH 5.04, 195 mM NaCl, 2 mM CaCl₂·2H₂O, conductivity 20.30 mS/cm at 22.4 °C) and five CV of re-equilibration buffer (50 mM Tris base pH 8.52, 100 mM NaCl, conductivity 12.01 mS/cm at 21.8 °C). A gradient elution was performed by mixing re-equilibration buffer with increasing volumes of elution buffer (50 mM Tris base pH 8.52, 100 mM NaCl, 100 mM CaCl₂·2H₂O, conductivity 29.44 mS/cm at 21.1 °C). The resin was regenerated using column regeneration buffer (50 mM Tris base, 2 M NaCl, pH 8.5, conductivity 156.3 mS/cm at 21.2 °C). After each experiment, the column was cleaned with three CV of 0.5 M NaOH and neutralized using five CV of re-equilibration buffer. The elution peak fraction of both purified samples was collected and concentrated using Amicon Ultra-10 centrifugal filter (10,000 MWCO, Merck) by centrifugation at 4000 r.c.f. for 15 min. Total protein concentration was measured at absorbance 280 nm using Lunatic (Unchained Labs), concentration was adjusted to 1 mg/mL, and samples were aliquoted in matrix tubes and stored at −80 °C for further analysis.

**Coomassie blue and western blotting**. Purified rFIX samples were separated by SDS–PAGE using a 4–12% Bis-Tris Bolt acrylamide gel (Invitrogen) in 1× MES running buffer (Invitrogen), under nonreducing conditions, following manufacturer's instructions. Protein bands were visualized either with colloidal Coomassie blue (Simply Blue; Invitrogen) or western blotting. For western blotting, proteins were transferred from gels onto a PVDF membrane (Bio-Rad) using a Trans-Blot® Turbo™ Transfer System Transfer Pack (Bio-Rad), following manufacturer's instructions. The membrane was blocked in 2% skim milk (Devondale) dissolved in phosphate-buffered saline solution (Sigma) with 0.05% Tween 20 (Sigma) for 30 min at room temperature. The following antibodies were used:

mouse IgG1 anti human FIX primary antibody 1C2 (AbCam, 1:1000 dilution) and HRP-conjugated anti mouse IgG (Bio-RAD, 1:1000 dilution). Membranes were developed with Novex ECL reagent (Invitrogen) and imaged using a Bio-Rad image workstation.

**Mass spectrometry sample preparation.** All samples were prepared as independent replicates: $N = 2$ for FIX characterization (DDA analyses) and $N = 3$ for FIX and FIX PTMs quantification (DIA analyses), and each sample was analyzed and measured once. Samples from purified rFIX and plasma-derived FIX (pdFIX; F0806, Sigma) were prepared by performing in-solution digestion, as previously described[131]. Briefly, triplicate 5 μg samples of purified rFIX and pdFIX were diluted in denaturation buffer (6 M guanidine hydrochloride, 50 mM Tris HCl buffer pH 8, and 10 mM dithiothreitol (DTT)) in a protein LoBind tube (Eppendorf) and incubated at 30 °C for 30 min in a MS100 Thermoshaker at 1500 r.p.m. Acrylamide was added to a final concentration of 25 mM and samples were incubated at 30 °C for 1 h in a Thermoshaker at 1500 r.p.m. Additional 5 mM DTT was added to quench acrylamide, and samples were precipitated with four volumes of 1:1 methanol:acetone and incubation at −20 °C for 16 h. Solvent was removed by two consecutive centrifugations at 21,000 r.c.f. for 10 min and 1 min, respectively, at room temperature. Samples were air dried at room temperature for ~15 min, resuspended in 50 mM ammonium bicarbonate solution containing one of the following proteases: 0.2 μg trypsin (T6567, Sigma), 0.17 μg GluC (11420399001, Sigma), 0.17 μg chymotrypsin (11418467001, Roche), and 0.04 μg AspN (19936721, Roche), and incubated at 37 °C for 16 h in a Thermoshaker at 1500 r.p.m. When followed by deglycosylation, all proteases were denatured by incubation at 95 °C for 5 min, and trypsin was also inactivated by addition of 1 mM phenylmethylsulfonyl fluoride. Samples were split into two protein LoBind tubes, one set of tubes was frozen at −20 °C, while 300 units of PNGase F (P0704S, New England Biolabs) was added to the other set of tubes and incubated at 37 °C for 16 h in a Thermoshaker at 1500 r.p.m. Samples were desalted with C18 ZipTips (ZTC18S960, Millipore).

Daily samples collected from bioreactors (day 1–13) were prepared with S-Trap columns following manufacturer's instructions (S-Trap C02-mini, Protifi), and as previously described[131]. Briefly, the supernatant samples were thawed on ice, centrifuged at 21,000 r.c.f. for 3 min at room temperature to remove aggregates and particulates, and 200 μL from each sample was transferred to protein LoBind tubes containing 200 μL of 2× solubilization buffer (10% SDS, 100 mM Tris HCl buffer pH 7.55, and 20 mM DTT) in triplicate. After incubation at 95 °C for 10 min, samples were cooled to room temperature before adding acrylamide to a final concentration of 25 mM. Samples were incubated at 30 °C for 1 h in a Thermoshaker at 1500 r.p.m., and acrylamide was quenched by adding additional 5 mM DTT. Samples were acidified by adding phosphoric acid to 1.2% (v/v) final concentration, diluted 1:7 with S-Trap binding buffer (90% methanol, 100 mM Tris HCl buffer pH 7.1), and loaded onto the S-Trap mini columns. The samples were washed four times with 400 μL of S-Trap binding buffer. Samples were then resuspended in 125 μL of 50 mM ammonium bicarbonate with 1 μg of trypsin, and columns were incubated at 37 °C for 15 h in a humidified chamber, without agitation. To recover the peptides, the columns were rehydrated with 80 μL of 50 mM ammonium bicarbonate, incubated at room temperature for 15 min, and centrifuged at 1000 r.c.f. for 1 min at room temperature. This was followed by subsequent elutions with 80 μL of 0.1% formic acid, followed by 80 μL of 50% acetonitrile in 0.1% formic acid. Elutions were pooled and desalted with C18 ZipTips.

**High pH reversed-phase peptide fractionation.** Peptides were fractionated by high pH reversed-phase fractionation essentially, as previously described[132]. Sep-Pak tC18 Vac 1cc (50 mg) cartridges (WAT054960, Waters) were equilibrated by washing twice with 100% acetonitrile followed by two washes with 0.1% formic acid. A total of 4 μL from each trypsinized bioreactor supernatant sample from each day were pooled, diluted in 0.1% formic acid to a final volume of 500 μL, and applied to the column. The peptides retained on the column were washed with milli-Q water and eluted in 9 × 500 μL fractions of increasing acetonitrile concentration (5–90% (v/v) acetonitrile in 0.1% triethylamine). All solutions applied to the cartridge were eluted by applying positive air pressure. Each fraction was collected in a protein LoBind tube, dried using a Genevac miVac centrifugal vacuum concentrator, and reconstituted in 100 μL of 0.1% formic acid.

**Methanolic HCl derivatization.** To detect γ-carboxyglutamic acid in peptides by positive ion mode LC-ESI-MS/MS, samples were chemically derivatized essentially as previously described[65,66], with some modifications. A total of 100 μL of day 13 bioreactor supernatant samples (concentrated in a 0.5 mL 10 kDa Amicon column) and 1 μg of purified rFIX and pdFIX were denatured, reduced, and precipitated overnight, as described above for in-solution digestion. Samples were completely dried in an acid resistant SpeedVac Savant SPD300DDA (Thermo Fisher), were resuspended in 25 μL of 3 M HCl in methanol (90964, Sigma), and incubated at 20 °C for 1 h. The methanol-HCl was removed by completely drying the samples in a SpeedVac Savant at 30 °C for 45 min followed by 40 °C for 20 min. Proteins were resuspended in 50 mM ammonium bicarbonate solution with 0.1 μg trypsin and incubated at 37 °C for 16 h in a Thermoshaker at 1500 r.p.m. Peptides were desalted with C18 ZipTips.

**Mass spectrometry analysis.** Desalted peptides were analyzed by LC-ESI-MS/MS using a Prominence nanoLC system (Shimadzu) and a TripleTof 5600 mass spectrometer with a Nanospray III interface (SCIEX) essentially as described[55,133,134]. Samples were desalted on an Agilent C18 trap (0.3 × 5 mm, 5 μm) at a flow rate of 30 μL/min for 3 min, followed by separation on a Vydac Everest C18 (300 Å, 5 μm, 150 mm × 150 μm) column at a flow rate of 1 μL/min. A gradient of 10–60% buffer B over 45 min where buffer A = 1 % ACN/0.1% FA and buffer B = 80% ACN / 0.1% FA was used to separate peptides. Gas and voltage settings were adjusted as required. An MS TOF scan across 350–1800 $m/z$ was performed for 0.5 s followed by DDA of up to 20 peptides with intensity >100 counts, across 100–1800 $m/z$ (0.05 s per spectra) using a collision energy (CE) of 40 ± 15 V. For DIA analyses (except for derivatized peptides, see below), MS scans across 350–1800 $m/z$ were performed (0.05 s), followed by high-sensitivity DIA mode with MS/MS scans across 50–1800 $m/z$, using 26 $m/z$ isolation windows for 0.1 s, across 400–1250 $m/z$. For DIA analysis of derivatized peptides, MS scans across 350–1800 $m/z$ were performed (0.05 s), followed by high-sensitivity DIA mode with MS/MS scans across 50–1800 $m/z$, using 6.2 $m/z$ isolation windows for 0.04 s, across 400–917 $m/z$. CE values for DIA samples were automatically assigned by Analyst TF1.7 (SCIEX) based on $m/z$ windows. The following samples were chosen for DDA analysis: one replicate per day for days 2, 5, 9, and 13 from each bioreactor, all the high pH fractionation samples, one replicate of each purified rFIX (from each bioreactor), and pdFIX digested with trypsin or trypsin and PNGase F. For DDA analysis of derivatized peptides, one replicate from each bioreactor were analyzed.

**Mass spectrometric characterization of rFIX.** To characterize the PTMs on rFIX, we analyzed the DDA data from purified rFIX and pdFIX digested with trypsin, GluC, AspN, or chymotrypsin, with or without N-glycan removal with PNGase F, using ProteinPilot v5.0.1 (SCIEX), and Preview and Byonic v2.13.17 (Protein Metrics). The following parameters were used in ProteinPilot: sample type, identification; cysteine alkylation, acrylamide; digestion, trypsin or GluC or chymotrypsin or AspN; instrument, TripleTOF 5600; ID focus, biological modifications; search effort, thorough; 1 % global FDR cutoff. The database used contained five isoforms of FIX (P00740-1, the UniProtKB canonical sequence with 461 amino acids (Fig. 1); the P00740-2 splice isoform, lacking the EGF-like 1 domain, amino acids 93–130; P00740-Δ1–46, lacking the signal sequence and propeptide, amino acids 1–46; P00740-T194A, P00740 containing the T$^{194}$A common natural variant[135,136]; and P00740-Q2G_P44V, rFIX Q$^2$G, P$^{44}$V used in this study), and a custom contaminants database from Proteome Discoverer (v2.0.0.802, Thermo Fisher Scientific). For some searches, the database contained only the P00740-1 and P00740-Q2G_P44V isoforms. The parameters used for searches in Byonic varied depending on the enzyme, and their full description can be found in Supplementary Data S5-Summary. The list of glycans searched in Byonic included the Byonic database containing 50 common biantennary N-glycans at Nglycan (rare1) for trypsin and GluC; or a modified N-glycan database combining the N-glycan 50 and 57 common biantennary glycoforms from Byonic and including 14 N-glycoforms with LacNAc extensions (LacNAc extensions are typical in CHO cells[137]; Supplementary Data S17) at Nglycan (i.e., at NXS/T; rare1) for AspN and chymotrypsin; and a list of O-glycan structures previously described for pdFIX (Supplementary Data S17) at Oglycan (i.e., at S/T; rare1 or rare2). The data for these Byonic searches are compiled in Supplementary Data S5 and the select Byonic peptide-spectrum matches (PSMs) for each PTM of interest can be found in Supplementary Data S6–S9.

**Label-free relative proteomic quantification.** To measure rFIX and the HCPs, we generated a combined ion library by searching all DDA files accompanying the SWATH/DIA experiments (high pH reversed-phase fractions, unfractionated samples from each bioreactor (obtained at days 2, 5, 9, and 13), purified FIX from each bioreactor, and pdFIX) in ProteinPilot v5.0.1 (SCIEX) using the following parameters: sample type, identification; Cys alkylation, acrylamide; digestion, trypsin; instrument, TripleTOF 5600; ID focus, biological modifications; search effort, thorough; 1 % global FDR cutoff (Supplementary Data S13). The database used included the entire *Cricetulus griseus* proteome UP000001075 (downloaded from Uniprot on 6 February 2019, total 24,194 proteins (23887 CHO K1 proteins, the five FIX isoforms described above, and a custom contaminants database from Proteome Discoverer). The V$^{108}$ATVSLPR$^{115}$ trypsin autolysis peptide (421.7584$^{2+}$ $m/z$) was manually added to this library with a confidence of 0.99 (Supplementary Data S14). Peptide abundance was measured by PeakView v2.2 (SCIEX) using the following parameters: proteins imported only <1% global FDR cutoff, shared peptides imported; six transitions/peptide; 99% peptide confidence threshold; 1% FDR; 2 min XIC extraction window; and 75 p.p.m. XIC width (Supplementary Data S15). The protein intensity output from PeakView was recalculated by eliminating the value of peptides measured with a FDR > 0.01 (Supplementary Data S15), using a modified version of our previously described script[138] (Supplementary Data S20). To calculate the relative changes in protein abundance through time in the bioreactor, the protein abundance data for each protein in each day was normalized to the abundance of trypsin in that sample, as previously described[54] (Supplementary Data S15). The mean ($N = 3$) for each trypsin-normalized protein abundance at each time point was log10 transformed and plotted as a heatmap, using GraphPad Prism v7.0a and 8.2.0 for Mac OS X

(GraphPad Software, San Diego, CA, USA). Statistical analysis of rFIX/trypsin values during bioreactor operation was performed using multiple *t* tests, with the two-stage linear setup procedure of Benjamini, Krieger, and Yekutieli to assign *P* values, and $Q = 1\%$, in GraphPad Prism. Statistical analyses of the HCPs and rFIX abundance in purified rFIX samples ($N = 3$) and at days 3, 6, 9, and 13 were performed using MSstats in R (refs. [55,133,138]; Supplementary Data S2). Volcano plots were made in GraphPad Prism.

**Quantification of posttranslational modifications**. To perform the relative quantification of PTMs in rFIX from bioreactor supernatant and purified samples, a PeakView ion library specific for FIX PTMs was generated. To do this, we first searched the DDA files corresponding to the SWATH/DIA-MS experiment from tryptic digests of purified rFIX and pdFIX, with and without PNGase F digestion, in Preview (v2.13.17, Protein Metrics), to obtain an overview of the PTMs and mass measurement accuracy. Next, to focus the search on the proteins present in the sample and increase search efficiency, a merged focused database was generated in Byonic by first performing two separate searches of one sample of purified rFIX from bioreactor H1 and one from H2 digested with trypsin and PNGase F, using the following parameters: fully specific cleavage at the C-terminus of Arg and Lys; 0 missed cleavages allowed; fragmentation type CID low energy; 20.0 p.p.m. precursor tolerance; 40.0 p.p.m. fragment tolerance; 1, 2, and 3 charges applied to charge-unassigned spectra; decoys and contaminants added; 1% protein FDR cutoff; allowing no modifications. The database used was the *C. griseus* proteome described above, and decoys and contaminants were included. The two focused databased were merged and the amino acid sequences of PNGase F and the rFIX variants P00740-Δ1–46 and P00740-Q2G_P44V were included to generate the merged focused database. We searched the DDA files in Byonic using the merged focused database, the error settings suggested by preview, and the following parameters: fully specific cleavage at the C-terminus of Arg and Lys; two missed cleavages allowed; fragmentation type QTOF/HCD; 15.0 p.p.m. precursor tolerance; 40.0 p.p.m. fragment tolerance; 1, 2, and 3 charges applied to charge-unassigned spectra; no decoys and contaminants added (as they were already included in the focused database); 1% protein FDR cutoff; and allowing four common and two rare modifications (described in Supplementary Data S18-Summary). To construct the ion library, we selected the best PSMs identified in Byonic for each manually validated posttranslationally modified tryptic peptide (Supplementary Data S18 and Supplementary Data S9). The criteria for selection of the best PSM included: PSM of a fully cleaved tryptic product, PSM within the highest scored PSMs (minimum of 100), highest coverage of *b* and *y* ions, presence of at least one Y ion and corresponding oxonium ions in the case of glycopeptides, and consistent RT across the different DDA files (see Supplementary Data S18-Notes, for more details). Modified peptides that shared a similar fragmentation pattern and fell in the same SWATH window at a similar RT were sorted by score, and the highest scoring manually validated PSM was included in the library (Supplementary Data S18-Selected PSMs). The MS/MS data, (including fragment ions, intensity, and RT) from the best PSMs was formatted into a PeakView ion library using an in-house written Python script[139] (Supplementary Data S12). Peptide abundance was quantified using the following PeakView settings: imported all proteins and shared peptides; measurement of all transitions/peptide and all peptides/protein; 99% peptide confidence threshold; 1% global FDR; 2 min XIC extraction window; 75 p.p.m. XIC width (Supplementary Data S3), and the peptide data was filtered out of values with FDR > 0.01 as above (Supplementary Data S3). Statistical analysis was performed using one-tailed *t* test in Excel (Microsoft).

The ion library for quantification of methylated γ-carboxylated peptides from rFIX was generated essentially as above, with the following changes. The parameters for Byonic searches are described in Supplementary Data S21-Summary. The criteria for selection of PSMs was the same as above (with select exceptions), and can be found in Supplementary Data S21-Notes. The MS/MS spectra of the PSMs selected to make the ion library can be found in Supplementary Data S10. The ion library generated (Supplementary Data S4) was manually modified to include the unfragmented precursors, as these were predominant ions in the DIA-MS/MS spectra for the selected PSMs, but not in the DDA-MS/MS spectra annotated by Byonic, and also to include Y ions with one methyl group to account for the potential methylation of the glycan in the TTEFWK methylated glycopeptide precursors identified containing one or two methyl groups (Y0 = 825.4141 *m/z*, Y1 = 1028.4935 *m/z*, and Y2 = 1190.5463 *m/z*). Peptide abundance was quantified in PeakView using the following settings: measurement of all peptides/protein and all transitions/peptide; 99% peptide confidence threshold; 1% FDR; 6 min XIC extraction window for all peptides except for the CSFEEAR variants, which used 2 min; and 75 p.p.m. XIC width (Supplementary Data S4). The quantified peptide data was filtered out of values with FDR > 0.01 (Supplementary Data S4). To calculate the level of carboxylation, the abundance of each variant peptide was first normalized to the abundance of all variations of the same peptide. The intensity of all peptide variants with 0, 1, 2, or 3 carboxyl groups were summed, and the % of each carboxyform was calculated. The data were plotted using GraphPad Prism. Statistical analysis was performed using one-tailed *t* test in Excel (Supplementary Data S4).

**Bioinformatic analysis**. Gene Ontology analysis was performed using ClueGO v2.5.4 (ref. [140]), a plugin of Cytoscape v.3.7.2 (ref. [141]), using as universe

background the CHO proteome UP000001075 described above formatted for ClueGO, and using standard settings.

**Statistics and reproducibility**. One bioreactor per condition was run. Technical replicate samples of bioreactor culture supernatant for each bioreactor were obtained daily for proteomic analysis of supernatant (15 mL), and at day 13 for proteomic analysis of purified product (2 L). Viability and metabolic measurements were performed once/day/bioreactor. For proteomic analysis, all samples were prepared as independent replicates: $N = 2$ for FIX characterization (DDA analyses) and $N = 3$ for FIX and FIX PTMs quantification (DIA analyses), and each sample was analyzed and measured once. Statistical analysis of rFIX/trypsin values during bioreactor operation was performed using multiple *t* tests, with the two-stage linear setup procedure of Benjamini, Krieger, and Yekutieli to assign *P* values, and $Q = 1\%$, in GraphPad Prism. Statistical analyses of the HCPs and rFIX abundance in purified rFIX samples ($N = 3$) and at days 3, 6, 9, and 13 in the supernatant were performed using MSstats in R (refs. [55,133,138]). Statistical analyses for the post-translationally modified peptides were performed using one-tailed *t* test in Excel (Microsoft). A $P < 0.05$ was considered significant for *t* test, and $P < 10^{-5}$ for MSstats.

**Reporting summary**. Further information on research design is available in the Nature Research Reporting Summary linked to this article.

## Data availability

The mass spectrometry raw data and ProteinPilot results are available in the ProteomeXchange Consortium (http://proteomecentral.proteomexchange.org)[142] via the PRIDE partner repository[143] with the dataset identifier PXD018229. The content of each uploaded file is described in Supplementary Data S22. A summary diagram of the proteomic workflows used in this work can be found in Supplementary Fig. S1. The results from the Byonic searches are compiled in Supplementary Data S6–S10. Original quantitative data and statistical tests are compiled in Supplementary Data S23 and S24. Any remaining information can be obtained from the corresponding author upon reasonable request.

## Code availability

All the software used in this study is either previously published and cited, commercially available, including Byonic (Protein Metrics), ProteinPilot and PeakView (SCIEX), and GraphPad Prism (GraphPad Software, San Diego, CA, USA), or can be found in Supplementary Data S20.

## Materials availability

The cell line used can be made available subjected to written permission from Lonza and a mutually agreed material transfer agreement.

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

## Acknowledgements

We thank Dr. Amanda Nouwens and Peter Josh from the Mass Spectrometry Facility at the School of Chemistry and Molecular Biosciences of The University of Queensland for their help and advice. This project utilized the scientific and technical assistance of the National Biologics Facility (NBF) at the University of Queensland. NBF is supported by Therapeutic Innovation Australia (TIA). TIA is supported by the Australian Government through the National Collaborative Research Infrastructure Strategy (NCRIS) program. L.F.Z. was funded by a Promoting Women Fellowship from the University of Queensland, Australia. B.L.S. was funded by an Australian National Health and Medical Research Council RD Wright Biomedical (CDF Level 2) Fellowship APP1087975. This work was funded by an Australian Research Council Discovery Project DP160102766 to B.L.S. and an Australian Research Council Industrial Transformation Training Centre IC160100027 to S.M.M., B.L.S., C.B.H., V.S., and Y.Y.L.

## Author contributions

L.F.Z., D.R.R., C.L.P., M.N., C.A., Y.Y.L., B.L.S., and C.B.H. designed the study. L.F.Z., D.R.R., and T.K.P. performed the experiments. L.F.Z., D.R.R., and B.L.S. wrote the manuscript. All authors edited and approved the final manuscript. L.F.Z., V.S., S.M.M., Y.Y.L., B.L.S., and C.B.H. obtained the funding.

## Competing interests

M.N., C.A., V.S., and Y.Y.L. are employees of CSL Ltd. The remaining authors declare no completing interests.
