## [Peer Review File · Communications Biology]

Reviewers' comments:

Reviewer #1 (Remarks to the Author):

Summary:

The author's intent is to attempt to bridge the gap between understanding the impact of process variables and determination of critical PTMs in a highly complex glycoprotein, Factor IX, which was synthesized from CHO cell hosts. To do so, a HRAM LC-MS/MS approach based on data independent acquisition was implemented which provided mass specific information and fragmentation data for all peptides, even with low signal intensities, detected post-chromatographic separation and MSMS detection.

Impression of the work:

The paper is reasonably well structured and written. The experimental section is well detailed however, due to the nature of the topic, there are several sample handling and data analysis steps described which potentially increase the margin for error if the aim was to reproduce the same experimental outcome. However, sufficient experimental details are provided by the authors in this regard and most of the samples handling methods described are quite typical of previously established peptide mapping workflows. The study has several value propositions in its favor such as simultaneous determination of FIX critical PTMs, HCPs, glycosylation and information on protein yield which provide sufficient originality and could be of real benefit for ensuring optimal process performance for FIX manufacturing. However, some revisions are recommended based on the comments below.

Specific comments:

1. One limitation of the study is that process understanding is limited to feed evaluation, which is proprietary in nature; hence while differences in process performance and product quality are observed, it is not possible to understand fully what drives them from the limited metabolite profile presented. It is understandable perhaps that the authors were primarily focused on varying process performance sufficiently to showcase the benefits of the DIA analytics approach. The experimental setup with cultures H1 and H2 described by the authors confirmed previous reports which indicated an inverse relationship between FIX yields and γ -carboxylation levels (Hallgren et. al, <https://doi.org/10.1021/bi026016e>). The authors attributed primarily to differences in culture feeds and consequent cell metabolism which as previously mentioned cannot be fully understood from the data available. Perhaps the authors could expand the introduction to add information on what upstream parameters are most likely to influence the critical PTMs of FIX based on previous knowledge.

2. The authors tend to over-use of the term "prediction" which is often used to describe a capability of the platform implemented to indeed predict critical product information, specifically quality, yield and impurity levels. While in some cases it might be appropriate, prediction especially in the context of correlating culture performance to product characteristics would generally require the development of mathematical models where early culture or early product quality data can be used to predict outcomes of the process and/or the product ultimate features. The results described fell at times slightly short in this regard, as stated below;

- Line 26 and Figure 2. The authors claim that the DIA based method can be used to predict yield from bioreactor cultures. Indeed, it is evident that signal intensity increases over time for both cultures H1 and H2 in what seems to be an exponential relationship but the authors don't provide further details on the actual daily concentration corresponding to the peak intensities reported.

Therefore it is not clear how actual FIX concentration can be predicted from the observed trends other than an approximate visual estimation. Vulcano plots are useful to illustrate significant changes but are not of real benefit for yield prediction.

- Figure 3. Data of γ -carboxylation levels is shown for purified and day 13 samples only. How soon in the culture can differences in γ -carboxylation between the two cultures be detected? Can the post-purification profile be predicted from the product quality profiles obtained within the early days of the culture? This is not clear from the text. Culture harvest was performed on day 13 therefore obtaining quality data just prior to harvest provides quite limited predictions on the profiles post-purification. How did γ -carboxylation levels differentiate for H1 and H2 cultures on day 9 for example when both cell viability values were above 90%? No significant differences in HCPs content were detected at this point which were subsequently found to impact the efficacy of the purification.

3. New PTMs were identified on purified FIX; how confident are the authors that these are not false positives and what steps were performed to minimize this risk?

- Line 385—No missed- cleavages allowed post-trypsin digestion; however, trypsin does not cut every lysine-arginine residues in the same way so there could be missed cleavages.
- Were the precursor and fragment mass tolerances optimized for PTM determination?

4. A critical evaluation of the limitations of peptide mapping methods would also be encouraged towards the end of the paper. For example, analytical profiling through the approach described requires multiple sample treatment steps which are laborious and time consuming. If the ultimate goal is product quality prediction during the process, such approaches would be difficult to implement with ongoing cultures. Can advancements in instrumentation capabilities facilitate the objectives set by the author? While the instrumentation used by the authors is capable of HRAM measurements at a resolution of approximately 30000 FWHM in the m/z range of interest, more advanced equipment like Q Exactive MS instruments can provide 3-4x times this resolution. Even reaching a stage where critical PTMs can be monitored from sub-unit analysis post-reduction would represent a significant time-saving benefit which could be more suitable for product quality predictions throughout cultures. Of course, this might not be possible for very complex proteins but instrumentation advancements might make this possible in the future.

Minor comments:

1. Line 255. Why add a lysis buffer? Technical replicate samples taken from days 1 to 13 were spun down and filtered through a 0.2 μm membrane (lines 178-180), therefore being cell free. Or are the samples mentioned in line 255 different daily samples?

2. Line 507. Was the detection of fully γ -carboxylated GLA peptides attempted in ESI negative mode? The signal intensity of such peptides might be low but the benefits of DIA could really be useful in this situation and potentially still provide fragmentation data.

3. Line 589-591. Is this hypothesis always valid irrespective of the purification method chosen? A comparison with some affinity based methods for example would have been beneficial.

4. Line 934-950 A further opportunity to evaluate different purification methods efficiencies through the DIA platform since clearly AEX purification does not ensure a totally pure sample as it is non-specific for FIX proteins.

5. Table S12--- The Y axis mentions that glucose for metabolite data is plotted but no curve is shown.

6. Table S12---Can glutamate profiles be also added for completion? There are significant differences in glutamine/ammonium trends and glutamate profiles can complement these profiles.

7. Table S16---A large number of HCPs were found in the H2 culture and not in H1. It is known that media composition and viability at harvest can impact the levels and type of HCPs. However, the vast majority of HCPs in H2 are classified as intra-cellular. Were the samples made cell free immediately upon collection from the bioreactors? This would be the typical approach in real-case scenarios hence most of the intracellular HCPs would not reach the purification step.

Reviewer #2 (Remarks to the Author):

Zacchi et al. provide a detailed account of the biotechnological production and quality assessment of coagulation factor IX (FIX) with a particular emphasis on the mass spectrometric characterization of FIX using data-independent acquisition (DIA) MS. FIX is a highly relevant biopharmaceutical for treatment of hemophilia and is an extensively post-translationally modified protein. The faithful preservation of the PTM pattern is therefore essential for the production of recombinant FIX (rFIX), while at the same time the production yields and purity (absence of host cell proteins) should be maximized. The present work gives a detailed insight into how biotechnological production processes can be optimized using modern mass spectrometry methods. DIA MS is used for the longitudinal profiling during 13 days of production in the bioreactor and for the analysis of the final purified product.

Specifically, the manuscript covers the following aspects of FIX production and characterization in a comparison of two bioreactor conditions:

- rFIX abundance differences in two bioreactors
- profile of gamma-carboxylation as one of the most important PTMs on FIX
- discovery and quantification of other known and new PTMs, including glycosylation and oxidation
- prediction of PTM profiles of rFIX from culture supernatant
- analysis of host cell protein profiles and content in the final product

The manuscript is very well written and I have hardly any specific comments. The authors provide an enormous level of detail in the main manuscript and in the supplementary information, which makes it a long read, but this clearly increases transparency. In summary, the manuscript provides an excellent example for the use of state-of-the-art mass spectrometry-based proteomics methods for biopharmaceutical production.

Minor comments:

Why was the coverage of "native", plasma-derived FIX so much lower for the GLA domain (line 642, Figure S5)? This is the only, massive difference in sequence coverage relative to rFIX. Could this be due to the presence of other, yet uncharacterized PTMs?

Reviewers' comments:

Reviewer #1 (Remarks to the Author):

Summary:

The author's intent is to attempt to bridge the gap between understanding the impact of process variables and determination of critical PTMs in a highly complex glycoprotein, Factor IX, which was synthesized from CHO cell hosts. To do so, a HRAM LC-MS/MS approach based on data independent acquisition was implemented which provided mass specific information and fragmentation data for all peptides, even with low signal intensities, detected post-chromatographic separation and MSMS detection.

Impression of the work:

The paper is reasonably well structured and written. The experimental section is well detailed however, due to the nature of the topic, there are several sample handling and data analysis steps described which potentially increase the margin for error if the aim was to reproduce the same experimental outcome. However, sufficient experimental details are provided by the authors in this regard and most of the samples handling methods described are quite typical of previously established peptide mapping workflows. The study has several value propositions in its favor such as simultaneous determination of FIX critical PTMs, HCPs, glycosylation and information on protein yield which provide sufficient originality and could be of real benefit for ensuring optimal process performance for FIX manufacturing. However, some revisions are recommended based on the comments below.

RESPONSE:

We thank the reviewer for their comments, and have responded in detail below.

Specific comments:

1. One limitation of the study is that process understanding is limited to feed evaluation, which is proprietary in nature; hence while differences in process performance and product quality are observed, it is not possible to understand fully what drives them from the limited metabolite profile presented. It is understandable perhaps that the authors were primarily focused on varying process performance sufficiently to showcase the benefits of the DIA analytics approach. The experimental setup with cultures H1 and H2 described by the authors confirmed previous reports which indicated a inverse relationship between FIX yields and γ -carboxylation levels (Hallgren et. al, <https://doi.org/10.1021/bi026016e>). The authors attributed primarily to differences in culture feeds and consequent cell metabolism which as previously mentioned cannot be fully understood from the data available. Perhaps the authors could expand the introduction to add information on what upstream parameters are most likely to influence the critical PTMs of FIX based on previous knowledge.

RESPONSE:

We thank the reviewer for this suggestion. We have expanded the introduction to include additional information on factors which can influence PTMs, both on FIX and in general:

L108-114

“The culture media and bioprocess operating conditions can also affect PTM occupancy and structure. Glycosylation profiles can be affected by glycoengineering strategies and the choice of cell line [42]; media composition including glucose, glutamine, lactate, ammonia, and amino acid concentrations [43-

45]; and process parameters such as dissolved oxygen, temperature, and pH [46, 47]. 'γ-Carboxylation depends on adequate cofactors, vitamin K, and processing enzymes, and an inverse relationship has been observed between the extent of 'γ-carboxylation and rFIX yield [48].”

2. The authors tend to over-use of the term “prediction” which is often used to describe a capability of the platform implemented to indeed predict critical product information, specifically quality, yield and impurity levels. While in some cases it might be appropriate, prediction especially in the context of correlating culture performance to product characteristics would generally require the development of mathematical models where early culture or early product quality data can be used to predict outcomes of the process and/or the product ultimate features. The results described fell at times slightly short in this regard, as stated below;

• Line 26 and Figure 2. The authors claim that the DIA based method can be used to predict yield from bioreactor cultures. Indeed, it is evident that signal intensity increases over time for both cultures H1 and H2 in what seems to be an exponential relationship but the authors don’t provide further details on the actual daily concentration corresponding to the peak intensities reported. Therefore it is not clear how actual FIX concentration can be predicted from the observed trends other than an approximate visual estimation. Vulcano plots are useful to illustrate significant changes but are not of real benefit for yield prediction.

RESPONSE:

The reviewer is correct, our MS/MS methods allow for relative quantification of FIX and its PTMs, as well as HCPs, but not for absolute quantification. Our methods could be easily adapted for absolute quantification, for example by including an isotope-labeled standard of known concentration during the sample preparation. This simple modification to the protocols would significantly increase the applicability of our methods. We thank the reviewer for pointing this out. We have revised the text to include this possibility on Line 1118-1123:

“For example, incorporating stable-isotope labelled standards during the sample preparation would allow absolute quantification of the biopharmaceutical product before and after purification [143]. Improvements in the sensitivity, speed, and automation of mass spectrometry instrumentation and sample preparation may also facilitate the application of these workflows for real-time industrial bioprocess and product monitoring.”

We have now added this point in the discussion. We have also exchanged “predict” for “correlate with” in line 31:

“Our methods provided a detailed overview of the dynamics of site-specific PTM occupancy and abundance on rFIX during production, which accurately correlated with the efficiency of purification and the quality of the purified product from different culture conditions.”

• Figure 3. Data of γ-carboxylation levels is shown for purified and day 13 samples only. How soon in the culture can differences in γ-carboxylation between the two cultures be detected? Can the post-purification profile be predicted from the product quality profiles obtained within the early days of the culture? This is not clear from the text. Culture harvest was performed on day 13 therefore obtaining quality data just prior to harvest provides quite limited predictions on the profiles post-purification. How did γ-carboxylation levels differentiate for H1 and H2 cultures on day 9 for example when both cell viability values were above 90%? No significant differences in HCPs content were detected at this point which were subsequently found to impact the efficacy of the purification.

We thank the reviewer for pointing out that this important feature of our method was not sufficiently clear in the text. Figure 2B shows data on γ -carboxylation levels measured directly in the raw bioreactor samples. Because of the difficulty in measuring fully γ -carboxylated GLA peptides discussed in the text (lanes 499-502, and please also see our response to reviewer 2 below), it was only possible to identify and measure undercarboxylated GLA peptides in H1 and H2 bioreactors, which we used as a proxy for incomplete carboxylation. Those measurements (Fig. 2B) show that already at \sim day 7 (or halfway into the bioreactor run) bioreactor H2 showed considerably less γ -carboxylation than H1, especially evident in the first GLA peptide, LEEFVQGNLER. Our inability to detect lack of γ -carboxylation earlier in the run could be due to low FIX abundance in the sample or to high level of γ -carboxylation in the FIX produced. Regardless, the results in Fig. 2B show that measurement of the raw sample during bioreactor operation several days before the run was terminated, and even when FIX levels and HCP levels were similar in both bioreactors, allowed for an early clear differentiation in the quality of the FIX produced in each bioreactor. We have clarified these points in the discussion by including additional text at L1019-1021:

“Importantly, our workflows can detect the onset of measurable differences in GLA LI-carboxylation (and other PTMs) during bioreactor operation (Fig. 2B), and this data can be correlated with bioreactor metabolic performance (Supplementary Fig. S2), and relative FIX yield (Fig. 1C).”

3. New PTMs were identified on purified FIX; how confident are the authors that these are not false positives and what steps were performed to minimize this risk?

RESPONSE:

All PTMs were identified with the help of Byonic and were then manually validated in the raw data. The PSMs chosen by Byonic are compiled in Supplementary information, and the Byonic output including the observed and calculated m/z and ppm error for all precursors assigned by Byonic are compiled in Supplementary Table S1. As examples, the new sulfation site at Tyr45 and the glycans in the EGF2 domain were reported in Figure 4B and 4C showing the manually annotated MS2 spectra, including observed precursor m/z value, Δ ppm.

• **Line 385—No missed- cleavages allowed post-trypsin digestion; however, trypsin does not cut every lysine-arginine residues in the same way so there could be missed cleavages.**

RESPONSE:

Line 385 describes the construction of a focused database using Byonic. The focused database is a subset of the CHO proteome that only includes the proteins that have been identified by the software (in this case, the proteins that accompany FIX in the sample). For the purposes of generating a focused database it is not critical to identify post-translational modifications, and so the search parameters were simplified as much as possible to expedite the search, including not allowing any missed cleavage (although this was not strictly necessary). We then used this focused database to narrow the search space to improve the efficiency of searches that involved multiple different PTMs at different sites with high PTM heterogeneity. For the searches for PTMs on FIX, 1 or 2 missed cleaves for trypsin were allowed, as shown in the “Summary” tab of Supplementary Tables S1 or S7, respectively. We have included more details on the parameters used for the searches in the methods text at L389-396:

“We searched the DDA files in Byonic using the merged focused database, the error settings suggested by Preview, and the following parameters: fully specific cleavage at the C-terminus of Arg and Lys; 2 missed cleavages allowed; fragmentation type QTOF/HCD; 15.0 ppm precursor tolerance; 40.0 ppm fragment tolerance; 1, 2, and 3 charges applied to charge-unassigned spectra; no decoys and contaminants added (as

they were already included in the focused database); 1% protein FDR cut-off; and allowing 4 common and 2 rare modifications (described in Supplementary Table S7-Summary).”

• **Were the precursor and fragment mass tolerances optimized for PTM determination?**

RESPONSE:

The precursor and fragment mass tolerances were determined prior to the searches using the software Preview (Protein Metrics), which accompanies Byonic, as described in the methods. This has been clarified by new text in the methods at L389-391:

“We searched the DDA files in Byonic using the merged focused database, the error settings suggested by Preview, and the following parameters:”

4. A critical evaluation of the limitations of peptide mapping methods would also be encouraged towards the end of the paper. For example, analytical profiling through the approach described requires multiple sample treatment steps which are laborious and time consuming. If the ultimate goal is product quality prediction during the process, such approaches would be difficult to implement with ongoing cultures. Can advancements in instrumentation capabilities facilitate the objectives set by the author? While the instrumentation used by the authors is capable of HRAM measurements at a resolution of approximately 30000 FWHM in the m/z range of interest, more advanced equipment like Q Exactive MS instruments can provide 3-4x times this resolution. Even reaching a stage where critical PTMs can be monitored from sub-unit analysis post-reduction would represent a significant time-saving benefit which could be more suitable for product quality predictions throughout cultures. Of course, this might not be possible for very complex proteins but instrumentation advancements might make this possible in the future.

RESPONSE:

This is an excellent point and we have addressed this by including additional text in the Discussion at L1120-1123:

“Improvements in the sensitivity, speed, and automation of mass spectrometry instrumentation and sample preparation may also facilitate the application of these workflows for real-time industrial bioprocess and product monitoring.”

Minor comments:

1. Line 255. Why add a lysis buffer? Technical replicate samples taken from days 1 to 13 were spun down and filtered through a 0.2 µm membrane (lines 178-180), therefore being cell free. Or are the samples mentioned in line 255 different daily samples?

RESPONSE:

To improve clarity, we have changed the name “lysis buffer” to “solubilization buffer” L256:

“2x solubilization buffer”

2. Line 507. Was the detection of fully γ-carboxylated GLA peptides attempted in ESI negative

mode? The signal intensity of such peptides might be low but the benefits of DIA could really be useful in this situation and potentially still provide fragmentation data.

RESPONSE:

We thank the reviewer for the suggestion. We did not implement any negative ESI mode analysis, but we acknowledge that it could be useful to identify and quantify the negatively charged GLA FIX peptides, as well as phosphorylated or sulfated precursors. We have added text to describe this possibility to the discussion at L1022-1024:

“While we performed all of our experiments in positive ion mode, negative ion mode DIA-MS could be an interesting alternative to identify and quantify highly negatively charged precursors, such as FIX GLA peptides [123, 124].”

3. Line 589-591. Is this hypothesis always valid irrespective of the purification method chosen? A comparison with some affinity-based methods for example would have been beneficial.

RESPONSE:

The anion exchange resin used in this work “enriches for 'y-carboxylated FIX due to the high negative charge of the 'y-carboxyglutamic acids in the GLA domain”. Purifying FIX with an affinity-based method that does not depend on the 'y-carboxylation levels would have most likely supported the hypothesis that lower 'y-carboxylation levels in the bioreactor translate to lower 'y-carboxylation levels post-purification. However, anion exchange resin is the state of the art for the first step in commercial FIX purification, and since the goal of this work was to develop a MS workflow that can be readily incorporated into the current bioprocessing workflows we chose to maintain the current purification standards. While very useful for optimizing downstream purification conditions, comparing the output of the POROS 50 HQ with other affinity-based methods would have been a out of the scope of this study. However, our workflows are also ideally suited to optimize downstream purification parameters, and we have added text to the discussion to emphasize this point at L1120-1023:

“Improvements in the sensitivity, speed, and automation of mass spectrometry instrumentation and sample preparation may also facilitate the application of these workflows for real-time industrial upstream and downstream bioprocess and product monitoring.”

Importantly, considering that we were using an enrichment step, the more reasonable hypothesis would have been that both H1 and H2 rFIX would be similarly carboxylated post-purification. We have also changed the text in the results to emphasize this unexpected result, which highlights the need to carefully monitor product quality both pre and post-purification, at L565-573:

“To purify rFIX we used a POROS 50 HQ strong anion-exchange resin, a quaternary polyethyleneimine that binds negatively charged molecules [49, 81]. POROS 50 HQ enriches for 'y-carboxylated FIX due to the high negative charge of the 'y-carboxyglutamic acids in the GLA domain [7, 50, 64]. We expected that purified rFIX from both bioreactors would be enriched in highly 'y-carboxylated forms compared to the bioreactor supernatant. Indeed, the extent of 'y-carboxylation in rFIX from bioreactor H1 and H2 was significantly higher after purification than in the culture supernatants (Fig. 3C-G and Supplementary Table S11-G). Although rFIX in bioreactor H1 supernatant was more 'y-carboxylated than in H2 (Fig. 3), we expected that purified H1 rFIX would have a similar level of 'y-carboxylation as H2 rFIX. However, while purified H1 rFIX was almost completely 'y-carboxylated, purified H2 rFIX was not (Fig. 3C-G and Supplementary Fig. S4).”

4. Line 934-950 A further opportunity to evaluate different purification methods efficiencies through the DIA platform since clearly AEX purification does not ensure a totally pure sample as it is nonspecific for FIX proteins.

RESPONSE:

We have added text to the discussion to emphasize this point at L1120-1123:

“Improvements in the sensitivity, speed, and automation of mass spectrometry instrumentation and sample preparation may also facilitate the application of these workflows for real-time industrial upstream and downstream bioprocess and product monitoring.”

5. Table S12--- The Y axis mentions that glucose for metabolite data is plotted but no curve is shown.

RESPONSE:

We thank the reviewer for noting this discrepancy. The data has now been included in Supplementary Table S12. The glucose profile for both bioreactors has also been added to Supplementary Figure S2, and the Y axis has been renamed.

Revised Supplementary Figure S2:

Supplementary Figure S2. Metabolic profile of CHO cells expressing rFIX in both fed batch conditions. CHO cells expressing rFIX and PACE/Furin were grown in fed batch bioreactor mode with either EfficientFeed A (H1, solid line and black circles) or EfficientFeed B (H2, dotted line and open square). The following metabolites were measured: (A) glutamine, (B) glutamate, (C) ammonium, (D) glucose, and (E) lactate.

6. Table S12---Can glutamate profiles be also added for completion? There are significantly differences in glutamine/ammonium trends and glutamate profiles can complement these profiles.

RESPONSE:

The data has now been included in Supplementary Table S12. The glutamate profile for both bioreactors has also been added to Supplementary Figure S2.

7. Table S16---A large number of HCPs were found in the H2 culture and not in H1. It is known that media composition and viability at harvest can impact the levels and type of HCPs . However, the vast majority of HCPs in H2 are classified as intra-cellular. Were the samples made cell free immediately upon collection from the bioreactors? This would be the typical approach in real-case scenarios hence most of the intracellular HCPs would not reach the purification step.

RESPONSE:

As described in the methods, cells were removed immediately upon collection by initial centrifugation and by filtering the sample, and so the increase in intracellular HCPs observed during bioreactor operation and post-purification does indicate cell lysis. The large number of HCPs found in the post-purification sample denotes a need for further purification steps, which are regularly performed during commercial FIX preparation, but were not performed in this work.

Reviewer #2 (Remarks to the Author):

Zacchi et al. provide a detailed account of the biotechnological production and quality assessment of coagulation factor IX (FIX) with a particular emphasis on the mass spectrometric characterization of FIX using data-independent acquisition (DIA) MS. FIX is a highly relevant biopharmaceutical for treatment of hemophilia and is an extensively post-translationally modified protein. The faithful preservation of the PTM pattern is therefore essential for the production of recombinant FIX (rFIX), while at the same time the production yields and purity (absence of host cell proteins) should be maximized. The present work gives a detailed insight into how biotechnological production processes can be optimized using modern mass spectrometry methods. DIA MS is used for the longitudinal profiling during 13 days of production in the bioreactor and for the analysis of the final purified product.

Specifically, the manuscript covers the following aspects of FIX production and characterization in a comparison of two bioreactor conditions:

- rFIX abundance differences in two bioreactors**
- profile of gamma-carboxylation as one of the most important PTMs on FIX**
- discovery and quantification of other known and new PTMs, including glycosylation and oxidation**
- prediction of PTM profiles of rFIX from culture supernatant**
- analysis of host cell protein profiles and content in the final product**

The manuscript is very well written and I have hardly any specific comments. The authors provide an enormous level of detail in the main manuscript and in the supplementary information, which makes it a long read, but this clearly increases transparency. In summary, the manuscript provides an excellent example for the use of state-of-the-art mass spectrometry-based proteomics methods for biopharmaceutical production.

RESPONSE:

We thank the reviewer for their comments.

Minor comments:

Why was the coverage of "native", plasma-derived FIX so much lower for the GLA domain (line 642, Figure S5)? This is the only, massive difference in sequence coverage relative to rFIX. Could this be due to the presence of other, yet uncharacterized PTMs?

RESPONSE:

It is indeed possible that there other as yet undescribed PTMs in the GLA domain of FIX. However, we do not believe this is the most likely explanation for the lack of detection of GLA peptides from plasma derived FIX. Instead, lack of detection of these peptides is more likely related to the high level of carboxylation in native FIX's GLA domain, compared to H1 and H2 FIX. As we suggest early in that section (Page 19, lines 508-511) "Fully γ -carboxylated GLA peptides are difficult to detect and identify in positive ion mode LC-ESI-MS/MS. Some of the reasons for this include the negative charge of the carboxyl groups, that γ -carboxylation appears to hinder protease cleavage, and neutral loss of CO₂ upon CID (collision induced dissociation) fragmentation [63, 64, 79, 80]." Therefore, the lack of detection of native GLA FIX is more likely related to the LC-MS/MS method and equipment than to additional PTMs in the GLA domain. We have included discussion of Reviewer's 1 suggestion of the possible utility of negative mode LC-MS to characterize and measure GLA peptides (see above), and have also added a sentence in line 642 for further clarification:

"The low coverage observed in pdFIX is most likely due to full γ -carboxylation of the GLA domain."

REVIEWERS' COMMENTS:

Reviewer #1 (Remarks to the Author):

The authors have covered extensively the original comments which were raised; as a result, the methodologies applied and their output will be clearer to potential readers.

In my opinion, the article is now worthy of publication in its latest state.